# Differentially Private Non-convex Learning for Multi-layer Neural Networks

## Abstract

This paper focuses on the problem of Differentially Private Stochastic Optimization for (multi-layer) fully connected neural networks with a single output node. In the first part, we examine cases with no hidden nodes, specifically focusing on Generalized Linear Models (GLMs). We investigate the well-specific model where the random noise possesses a zero mean, and the link function is both bounded and Lipschitz continuous. We propose several algorithms, and our analysis demonstrates the feasibility of achieving an excess population risk that remains invariant to the data dimension. We also delve into the scenario involving the ReLU link function, and our findings mirror those of the bounded link function. We conclude this section by contrasting well-specified and misspecified models, using ReLU regression as a representative example. In the second part of the paper, we extend our ideas to two-layer neural networks with sigmoid or ReLU activation functions in the well-specified model. In the third part, we study the theoretical guarantees of DP-SGD in Abadi et al. (2016) for fully connected multi-layer neural networks. By utilizing recent advances in Neural Tangent Kernel theory, we provide the first excess population risk when both the sample size and the width of the network are sufficiently large. Additionally, we discuss the role of some parameters in DP-SGD regarding their utility, both theoretically and empirically.

## 1 Introduction

In the domain of machine learning, extracting knowledge from data harboring sensitive attributes is an evolving concern. Such a task mandates algorithms that can proficiently interpret the data while upholding established privacy benchmarks. Differential privacy (DP) Dwork et al. (2006), in this context, has gained traction as a seminal framework for statistical data protection. Recognized widely in contemporary research, DP ensures that individual data remains non-retrievable post-analysis, offering a robust defense mechanism against privacy infractions. This underscores a burgeoning interest in devising learning architectures where DP considerations are intrinsically woven into the analytic process.

Stochastic Optimization (SO) and its empirical form, Empirical Risk Minimization (ERM), are the most fundamental models in machine learning and statistics. They have numerous applications in fields such as medicine, finance, genomics, and social science. However, these applications often involve sensitive data, making it essential to design differentially private algorithms for SO and ERM, corresponding to the problems of DP-SO and DP-ERM, respectively. While DP-SO and DP-ERM have been extensively studied for more than a decade, most of the existing work considers the case where the loss function is convex. The problem of DP-SO and DP-ERM with non-convex loss functions remains far from well-understood due to their complex nature. Although there is some preliminary work, such as Wang & Xu (2019); Wang et al. (2019a;b); Song et al. (2021), there are still two critical issues. Firstly, most of the existing work adopts the gradient norm of the population risk function to measure the utility, which is quite different from the convex case where we use the excess population risk instead. However, using the gradient norm is inadequate for indicating how close the private model is to the optimal solution Agarwal et al. (2017). Secondly, while recently there has been some work considering the excess population risk for non-convex loss functions Wang et al. (2019a), most research has narrowly focused on general non-convex loss functions, overlooking the intricacies of neural network structures. To address these issues, this paper provides the first comprehensive and theoretical study

of DP Fully Connected Neural Networks (with a single output node) and presents several bounds of excess population risk. Specifically, our contributions can be summarized as follows:

1. In the first part of the paper, we focus on the simplest neural network structure: neural networks without hidden nodes, aptly referred to as non-convex Generalized Linear Models (GLMs). We first address the well-specified model that is characterized by zero-mean random noise, combined with bounded and Lipschitz link functions. For this setup, we introduce an $(\epsilon, \delta)$-DP algorithm and demonstrate its efficacy with an output upper bound $\tilde{O}(\frac{1}{\sqrt{n}} + \min\{\frac{1}{(n\epsilon)^{\frac{2}{3}}}, \frac{\sqrt{\theta}}{n\epsilon}\})$. Here $\theta$ is an upper bound on the expected rank of the data matrix and $n$ is the sample size. We then broaden our study to cases with unbounded link functions, specifically when employing the ReLU activation function. In this scenario, we establish that an upper bound of $\tilde{O}(\frac{1}{\sqrt{n}} + \min\{\frac{\sqrt{d}}{n\epsilon}, \frac{1}{(n\epsilon)^{\frac{2}{3}}}\})$ is feasible. Subsequently, our attention pivots to the misspecified model. To delineate its nuances vis-à-vis the well-specified model, we spotlight the ReLU activation function as a representative case. Within this scope, we innovate a distinct version of DP Gradient Descent, showcasing a sample complexity of $\tilde{O}(\max\{\frac{\sqrt{d}}{\epsilon\alpha}, \frac{d}{\alpha^2}\})$. This sample complexity guarantees that the difference between the population risk of our private estimator and $c \cdot \text{opt}$ is no more than $\alpha$, where opt is the optimal value of population risk and $c > 0$ is some constant.

2. Next, we extend our ideas to the problem of privately learning two-layer neural networks. Specifically, we consider the well-specified model and study the cases where the activation functions are either sigmoid or ReLU. Our main contribution is to establish the sample complexity required to achieve an error of $\alpha$ for excess population risk. For the sigmoid case, we show that the sample complexity is $O((\frac{kC_1}{\alpha})^{2C_1}\frac{1}{\epsilon^2})$, where $k$ is the number of hidden nodes and $C_1$ is a positive constant. For the ReLU case with $k$ hidden nodes, we show that the sample complexity is $O(4^{C_2\frac{k}{\alpha}}\frac{1}{\epsilon^2})$, where $C_2$ is a positive constant.

3. In the last part, we consider general multi-layer fully connected neural networks. Rather than introducing new methods, we delve into the theoretical guarantees of the standard DP-SGD as detailed in Abadi et al. (2016). Drawing upon recent advancements in the Neural Tangent Kernel (NTK), we present the inaugural excess population risk bound for networks where both the width of each layer and the sample size are sufficiently large. In essence, this bound is composed of three elements: an approximation error attributable to NTK, an error arising from the Gaussian noise introduced in every iteration, and a combined term representing the convergence rate and sampling error. Building on our theoretical framework, we then delve into the intricate interplay and trade-offs between various parameters. We also provide experimental studies to corroborate our theoretical findings.

## 2 Related Work

As we mentioned earlier, there is a long list of work on DP-SO and DP-ERM. Thus, here we only mention the theoretical work that is close to ours.

**Private non-convex learning.** In DP-SCO/DP-ERM with convex loss functions, the excess population risk is commonly used to measure the utility. However, in the non-convex case, there are three general ways to measure the utility. The first approach is based on the first-order stationary condition, such as the gradient $\ell_2$-norm of the population risk function Wang & Xu (2019); Song et al. (2021); Zhou et al. (2020); Bassily et al. (2021); Zhang et al. (2021). However, there are some issues with this measure. Firstly, previous work has shown that the gradient norm tends to 0 as the sample size $n$ goes to infinity, but there is no guarantee that such a private estimator will be close to any non-degenerate local minimum Agarwal et al. (2017). Secondly, the gradient-norm estimator is not always consistent with the excess empirical (population) risk of the loss function Wang et al. (2019a).

The second approach considers using the second-order stationary condition as the measure, which involves considering both the norm of the gradient and the Hessian matrix minimal eigenvalue of the population risk function Wang et al. (2019a); Wang & Xu (2020a). The motivation for this approach is based on the fact that for many machine learning problems, such as matrix completion and dictionary learning, any second-order stationary point is a local minimum of the problem, and all the local minima are the global minimum. Thus, finding a global minimum is equivalent to finding a second-order stationary point. However, the main

disadvantage of this measure is that it is only reasonable for some problems, and it is unknown whether general neural networks satisfy the above property.

The third approach is to directly use the excess population risk, which is similar to the convex case, and our work goes in this direction. However, most of the previous work only considers some specific class of loss functions, such as Polyak-Lojasiewicz loss Wang et al. (2017). Wang et al. (2019a) provided the first study of DP-ERM with general non-convex loss, but their bound is $O(\frac{d}{(\log n)\epsilon})$, which is quite large. Compared to their results, our work considers general neural networks and provides improved bounds.

**DP-GLM.** DP-SO/DP-ERM with Generalized Linear loss (DP-GLL) and DP-GLM have received considerable attention in recent years. For convex loss functions, Jain & Thakurta (2014) provided the first study on DP-GLL and showed that in the unconstrained case, the error bound can achieve $\tilde{O}(\frac{1}{\sqrt{n}\epsilon})$ in general, which is quite different from the bound $O(\frac{\sqrt{d}}{n\epsilon})$ for general convex DP-ERM. Later, Kasiviswanathan & Jin (2016) studied the same problem and showed that in the constrained case, the error bound could only depend on the Gaussian width of the underlying constraint set. For the unconstrained setting, Song et al. (2021) showed an improved bound of $O(\frac{\sqrt{\theta}}{n\epsilon})$, where $\theta$ is the rank of the expectation of the data matrix. For constrained DP-GLM, Bassily et al. (2021) considered various settings where the loss could be smooth/non-smooth and in the $\ell_p$ space for general $1 \le p \le 2$. Recently, Arora et al. (2022b) studied the optimal rates of DP-GLM in the unconstrained setting. Specifically, when the loss is smooth and non-negative but not necessarily Lipschitz, it showed the optimal rate of $\tilde{O}(\frac{1}{\sqrt{n}} + \min\{\frac{1}{(n\epsilon)^{2/3}}, \frac{\sqrt{d}}{n\epsilon}\})$. When the loss is Lipschitz, the optimal rate is $\tilde{O}(\frac{1}{\sqrt{n}} + \min\{\frac{1}{\sqrt{n\epsilon}}, \frac{\sqrt{\theta}}{n\epsilon}\})$. For non-convex losses, Song et al. (2021); Bassily et al. (2021) provided bounds that are independent of the dimension for the gradient $\ell_2$-norm of the population risk function. Wang et al. (2019a); Hu et al. (2022) studied the excess population risk for some specific GLMs and showed that their bound can be only logarithmic in the dimension. However, they need to assume the constraint set is an $\ell_1$-norm ball, while our work does not require such an assumption.

## 3 Preliminaries

**Definition 1** (DP Dwork et al. (2006)). Given a data universe $\mathcal{X}$, we say that two datasets $D, D' \subseteq \mathcal{X}$ are neighbors if they differ by only one data record, which is denoted as $D \sim D'$. A randomized algorithm $\mathcal{A}$ is $(\epsilon, \delta)$-differentially private (DP) if for all neighboring datasets $D, D'$ and for all events $S$ in the output space of $\mathcal{A}$, we have

$$\Pr(\mathcal{A}(D) \in S) \le e^\epsilon \Pr(\mathcal{A}(D') \in S) + \delta.$$

**Lemma 1** (Gaussian Mechanism). *Given any function $q : \mathcal{X}^n \to \mathbb{R}^d$, the Gaussian mechanism is defined as $q(D) + \xi$ where $\xi \sim \mathcal{N}(0, \frac{2\Delta_2^2(q)\log(1.25/\delta)}{\epsilon^2}\mathbb{I}_d)$, where $\Delta_2(q)$ is the $\ell_2$-sensitivity of the function $q$, i.e., $\Delta_2(q) = \sup_{D \sim D'} ||q(D) - q(D')||_2$. Gaussian mechanism preserves $(\epsilon, \delta)$-DP for $0 < \epsilon, \delta < 1$.*

**Definition 2** (DP-SO Bassily et al. (2014)). Given a dataset $D = \{z_1, \cdots, z_n\}$ from a data universe $\mathcal{Z}$ where each $z_i = (x_i, y_i)$ with a feature vector $x_i$ and a label/response $y_i$ is i.i.d. sampled from some unknown distribution $\mathcal{P}$, a convex constraint set $\mathcal{W} \subseteq \mathbb{R}^d$, and a (non-convex) loss function $\ell : \mathcal{W} \times \mathcal{Z} \mapsto \mathbb{R}$. Differentially Private Stochastic Optimization (DP-SO) is to find a model $w^{\text{priv}}$ to minimize the population risk, *i.e.*, $L_\mathcal{P}(w) = \mathbb{E}_{(x,y)\sim\mathcal{P}}[\ell(w; x, y)]$ with the guarantee of being differentially private.[1] The utility of $w^{\text{priv}}$ is measured by the (expected) excess population risk $\mathbb{E}L_\mathcal{P}(w^{\text{priv}}) - \min_{w\in\mathcal{W}} L_\mathcal{P}(w)$, where the expectation takes over the randomness of the algorithm and the input data. Besides the population risk, we can also measure the *empirical risk* of dataset $D$: $\hat{L}(w, D) = \frac{1}{n}\sum_{i=1}^n \ell(w, z_i)$.

It is notable that besides the error bound, to better demonstrate our results, we may also consider the sample complexity to achieve a fixed error $\alpha$ to measure the utility of DP algorithms.

**Definition 3.** A function $f(\cdot)$ is $G$-Lipschitz if for all $w, w' \in \mathcal{W}$, $|f(w) - f(w')| \le G\|w - w'\|_2$.

**Definition 4.** A function $f(\cdot)$ is $\beta$-smooth on $\mathcal{W}$ if for all $w, w' \in \mathcal{W}$, $f(w') \le f(w) + \langle \nabla f(w), w' - w \rangle + \frac{\beta}{2}\|w' - w\|_2^2$.

---

[1]Note that in this paper, we consider the improper learning case, that is $w^{\text{priv}}$ may not be in $\mathcal{W}$.

**Definition 5.** A function $f(\cdot)$ is $\alpha$-strongly convex on $\mathcal{W}$ if for all $w, w' \in \mathcal{W}$, $f(w') \geq f(w) + \langle \nabla f(w), w' - w \rangle + \frac{\alpha}{2} \|w' - w\|_2^2$.

**Definition 6.** A random matrix $\Phi \in \mathbb{R}^{k \times d}$ satisfies $(\alpha, \beta)$-Johnson-Lindentrauss (JL) property if for any $u, v \in \mathbb{R}^d$ and any $\alpha > 0$ we have $\mathbb{P}[|\langle \Phi u, \Phi v \rangle - \langle u, v \rangle| > \alpha \|u\|_2 \|v\|_2] \leq \beta$, where the probability takes over the randomness of the distribution of $\Phi$.

Specifically, when $R \in \mathbb{R}^{k \times d}$ is a random Gaussian matrix with $k = O(\frac{\log 1/\beta}{\alpha^2})$ each entry is i.i.d. sampled from $\mathcal{N}(0, 1)$. Then the matrix $A = \frac{1}{\sqrt{k}} R$ satisfies $(\alpha, \beta)$-JL property.

## 4 Private Non-convex GLMs

### 4.1 Well-specified Model

#### 4.1.1 Bounded Link Function Case

In this section, we will examine the problem of Generalized Linear Models (GLMs), which are neural networks without hidden layers and with a single output neuron. Specifically, we will begin by considering a simplified scenario in which the statistical model is well-specified,[2] meaning that the Bayes optimal classifier satisfies $\mathbb{E}[y|x] = \sigma(\langle w^*, x \rangle)$ for some underlying parameter $w^* \in \mathbb{R}^d$ and non-convex link function $\sigma$:

$$y = \sigma(\langle w^*, x \rangle) + \zeta, \tag{1}$$

where $\zeta$ is random noise with zero mean. In the following, we will introduce several assumptions that will be used throughout this section.

**Assumption 1.** Assume there exist constants $W, G, B = O(1)$ such that $\|w^*\|_2 \leq W$, $y \in [-B, B]$ and the link function $\sigma : \mathbb{R} \mapsto [-B, B]$ is $G$-Lipschitz and non-monotone decreasing. We also assume $\|x\|_2 \leq 1$. [3]

The assumption of $\|w^*\|_2 \leq W$ for a given known $W$ is a recurring theme in the literature on private learning and statistical estimation. Notably, even in linear models where $\sigma$ serves as the identity function, this presumption consistently appears in prior research Cai et al. (2021); Wang & Xu (2020b).

In fact, many activation functions that are commonly used in neural networks satisfy Assumption 1, such as sigmoid function $\sigma(x) = \frac{1}{1 + \exp(-x)}$ and tanh function $\sigma(x) = \frac{\exp(x) - \exp(-x)}{\exp(x) + \exp(-x)}$.

Under the well-specified model (1), we consider the expected squared error as the population risk function, i.e., $L_{\mathcal{P}}(w) = \mathbb{E}_{(x,y) \sim \mathcal{P}} (\sigma(\langle w, x \rangle) - y)^2$.

To solve the problem, the most natural idea is to approximate the population function $L_{\mathcal{P}}(w)$ by some convex stochastic function. Motivated by Kalai & Sastry (2009); Kakade et al. (2011), here we consider the following surrogate (convex) loss function:

$$\ell(w; x, y) = \int_0^{\langle w, x \rangle} (\sigma(z) - y) dz. \tag{2}$$

The following result shows that the loss $\ell$ is convex, Lipschitz and smooth:

**Lemma 2.** *Under Assumption 1, for any $(x, y) \sim \mathcal{P}$, function $\ell(\cdot; x, y)$ is convex and $2B$-Lipschitz. Moreover, if $\sigma$ has (sub)gradient anywhere, then the rank of the Hessian matrix for $\ell(\cdot; x, y)$ is 1, and $\ell(\cdot; x, y)$ is $G$-smooth.*

In the following, we use the notations $L^{\ell}(w; D) = \frac{1}{n} \sum_{i=1}^n \ell(w; x_i, y_i)$ and $L_{\mathcal{P}}^{\ell}(w) = \mathbb{E}[\ell(w; x, y)]$ to represent the empirical risk and population risk functions for the loss $\ell$ in (2), respectively. The following lemma, given by Kakade et al. (2011), shows that the optimal parameter $w^*$ is also the minimizer of $L_{\mathcal{P}}^{\ell}(\cdot)$. Moreover, for any $w$, the excess population risk of $w$ is dominated by the excess population risk of loss $\ell(w; x, y)$.

---

[2]In the literature, the well-specified setting is also extensively referred to as the "noisy teacher" setting Frei et al. (2020) or the well-structured noise model Goel & Klivans (2019)

[3]For simplicity, here we assume $\|x\|_2 \leq 1$, and for the range of $\sigma$ we use the same $B$ as the range of $y$, we can easily extend our results to general cases.

---

**Algorithm 1** DP non-convex GLM

---

**Require:** Private dataset: $D = \{(x_i, y_i)\}_{i=1}^n$, link function $\sigma$ satisfying Assumption 1 and has (sub)gradient anywhere; privacy parameters $0 < \epsilon, \delta \leq 1$, upper bound $\theta$ of the expected rank of data matrix.

1: If $\epsilon > \frac{\theta^{\frac{3}{2}}}{n}$, run Algorithm 2. Otherwise, run Algorithm 3.

---

**Lemma 3.** *For any $w \in \mathbb{R}^d$, we have $L_{\mathcal{P}}(w) - L_{\mathcal{P}}(w^*) \leq 2G(L_{\mathcal{P}}^\ell(w) - L_{\mathcal{P}}^\ell(w^*))$.*

Thus, motivated by Lemma 3, now we aim to find a private estimator $w^{priv} \in \mathbb{R}^d$ to minimize $L_{\mathcal{P}}^\ell(w)$. Moreover, we can see from the form of the loss $\ell$ and Lemma 2 that if $\sigma$ has subgradient anywhere, then $\ell(\cdot)$ will be a generalized linear loss, i.e., $\ell(w; x, y) = g(\langle w, x \rangle, y)$ where $g(\cdot, y)$ is convex, $2B$-Lipschitz, and $G$-smooth. Therefore, we can use unconstrained DP-SO algorithms for convex generalized linear loss to $\ell$ to obtain a private estimator. Here, we adopt the Phased SGD method for convex GLM in Bassily et al. (2021) (see Algorithm 2 for details). Furthermore, motivated by Arora et al. (2022b), we propose a new method that uses a JL matrix to preprocess the data and then performs Algorithm 2 over the projected data. Note that using Phased SGD is crucial for our analysis of two-layer neural networks in later sections (see Remark 9 for details). The entire algorithm is provided in Algorithm 1.

---

**Algorithm 2** Phased SGD for non-convex GLM

---

**Require:** Private dataset: $D = \{(x_i, y_i)\}_{i=1}^n$, link function $\sigma$ satisfying Assumption 1 and has (sub)gradient anywhere; privacy parameters $0 < \epsilon, \delta \leq 1$.

1: Denote the loss function $\ell$ as $\ell(w; x, y) = \int_0^{\langle w, x \rangle} (\sigma(z) - y) dz$.

2: Set $k = \lceil \log_2(n) \rceil$, partite the whole dataset S into $k$ subsets $\{D_1, \cdots, D_k\}$. Denote $n_i$ as the number of samples in $D_i$, i.e., $|D_i| = n_i$ where $n_i = \lfloor 2^{-i} n \rfloor$. Take a random initial vector $w_0 \in \mathcal{W}$.

3: **for** $i = 1, \cdots, k$ **do**

4:     Let $\eta_i = \frac{\eta}{4^i}$ and $w_i^1 = w_{i-1}$.

5:     **for** $t = 1, \cdots, n_i$ **do**

6:         Update $w_i^{t+1} = w_i^t - \eta_i \nabla \ell(w_i^t; x_i^t, y_i^t) = w_i^t - \eta_i(\sigma(\langle w_i^t, x_i^t \rangle) - y_i^t) x_i^t$, where $(x_i^t, y_i^t)$ is the $t$-th sample of $D_i$.

7:     **end for**

8:     Denote $w_i = \bar{w}_i + \zeta_i$, where $\bar{w}_i = \frac{1}{n_i} \sum_{t=1}^{n_i} w_i^t$ and $\zeta_i \sim \mathcal{N}(0, \tau_i^2 \mathbb{I}_d)$ with $\tau_i = \frac{8B\eta_i \sqrt{\log \frac{1}{\delta}}}{\epsilon}$.

9: **end for**

---

In the following, we will show the utility. We denote $\theta$ as the upper bound of $\mathbb{E}_{D \sim \mathcal{P}^n}[\text{Rank}(V)]$, where $V$ is a matrix whose columns are an eigenbasis for $\sum_{i=1}^n x_i x_i^T$. Note that we always have $\mathbb{E}_{D \sim \mathcal{P}^n}[\text{Rank}(V)] \leq n$.

**Theorem 1.** *Under Assumption 1 and if $\sigma$ has (sub)gradient anywhere, for any $0 < \epsilon, \delta < 1$, Algorithm 2 is $(\epsilon, \delta)$-DP. Moreover, when $\eta = O(\min\{\frac{\epsilon}{\sqrt{\theta \log \frac{1}{\delta}}}, \frac{1}{\sqrt{n}}\}) \leq \frac{2}{G}$, we have*

$$\mathbb{E}L_{\mathcal{P}}(w_k) - L_{\mathcal{P}}(w^*) \leq O\left(\frac{1}{\sqrt{n}} + \frac{\sqrt{\theta \log \frac{1}{\delta}}}{n\epsilon}\right).$$

**Theorem 2.** *Under Assumption 1 and if $\sigma$ has (sub)gradient everywhere, let $m = O(\log(n/\delta)(n\epsilon)^{\frac{2}{3}})$, then Algorithm 3 is $(\epsilon, \delta)$-DP for any $0 < \epsilon, \delta < \frac{1}{2G}$. Moreover, when $\eta = O(\min\{\frac{\epsilon}{\sqrt{m \log \frac{1}{\delta}}}, \frac{1}{\sqrt{n}}\}) \leq 1$, we have*

$$\mathbb{E}L_{\mathcal{P}}(\hat{w}) - L_{\mathcal{P}}(w^*) \leq \tilde{O}\left(\frac{\sqrt{\log \frac{1}{\delta}}}{(n\epsilon)^{\frac{2}{3}}} + \frac{1}{\sqrt{n}}\right),$$

*where the Big-$\tilde{O}$ notation omits the term of $\log(n/\delta)$.*

*Remark* 3. The output of Algorithm 1 achieves an error of $\tilde{O}(n^{-\frac{1}{2}} + \min\{\sqrt{\theta}(n\epsilon)^{-1}, (n\epsilon)^{-\frac{2}{3}}\})$. This rate appears to be better than the lower bound for DP convex and Lipschitz Generalized Linear loss (DP-GLL)

---

Arora et al. (2022b), which is near-optimal at $O(n^{-\frac{1}{2}} + \min\{\sqrt{\theta}(n\epsilon)^{-1}, (n\epsilon)^{-\frac{1}{2}}\})$. However, these results are not contradictory, as Arora et al. (2022b) considers a more general class of loss functions. In fact, the above lower bound for DP-GLL only holds for the case where $L_{\mathcal{P}}(w) = \mathbb{E}[|y - \langle w, x \rangle|]$, while our problem mainly focuses on the squared loss $L_{\mathcal{P}}(w) = \mathbb{E}[(y - \sigma(\langle w, x \rangle))^2]$. Therefore, the lower bound does not apply to our problem. Additionally, Arora et al. (2022b) considers the smooth and non-negative generalized linear loss, which is not necessarily Lipschitz, and shows that the near-optimal rate is $\tilde{O}(n^{-\frac{1}{2}} + \min\{(n\epsilon)^{-\frac{2}{3}}, \sqrt{d}(n\epsilon)^{-1}\})$. Again, these results are not contradictory to ours.

---

**Algorithm 3** DP-Projected Phased SGD for non-convex GLM

---

**Require:** Private dataset: $D = \{(x_i, y_i)\}_{i=1}^n$, link function $\sigma$ satisfying Assumption 1 and has (sub)gradient anywhere; privacy parameters $0 < \epsilon, \delta \leq 1$.
1: Sample a JL matrix $\Phi \in \mathbb{R}^{m \times d}$, and denote $\tilde{D} = \{(\Phi x_1, y_1), \cdots, (\Phi x_n, y_n)\}$.
2: Run Algorithm 2 on the projected dataset $\tilde{D}$, i.e., in Line 6 of Algorithm 2, update

$$w_i^{t+1} = w_i^t - \eta_i(\sigma(\langle w_i^t, \tilde{x}_i^t \rangle) - y_i^t)\tilde{x}_i^t,$$

where $(\tilde{x}_i^t, y_i^t)$ is the $t$-th sample of $\tilde{D}_i$. And in Line 8, $\zeta_i \sim \mathcal{N}(0, \tau_i^2 \mathbb{I}_m)$ with $\tau_i = \frac{16B\eta_i \sqrt{\log \frac{2}{\delta}}}{\epsilon}$. Denote the output as $w_k$.
3:
4: Return $\hat{w} = \Phi^T w_k$

---

### 4.1.2 More General Link Functions

One issue with the previous approach is that the link function $\sigma$ should have a subgradient everywhere so that the surrogate function $\ell(\cdot; x, y)$ is smooth by Lemma 2. However, unlike the convex case, this assumption may not always hold since some non-convex functions may have no subgradient at some point. We will address this case and demonstrate that it is possible to achieve the same bounds as in Theorem 1 and Theorem 2 (but with higher time complexity).

Since the surrogate loss function, in this case, becomes non-smooth, the issue lies in finding a method to make it smooth. To illustrate our approach, we first recall the Moreau envelope smoothing technique that can be used to make a non-smooth function smooth Moreau (1965). Let $\mathcal{M}$ be a (potentially unbounded) closed interval, $y \in \mathbb{R}$ and $\beta > 0$. Consider a function $\ell : \mathcal{M} \mapsto \mathbb{R}$. The $\beta$-Moreau envelop of $\ell$ is defined as

$$\ell_\beta(x) = \min_{u \in \mathcal{M}}[\ell(u) + \frac{\beta}{2}|u - x|^2].$$

Denote the proximal operator with respect to $\ell$ as

$$\text{prox}_\ell^\beta(x) = \arg \min_{u \in \mathcal{M}}[\ell(u) + \frac{\beta}{2}|u - m|^2].$$

If $\ell$ is a convex function, then its Moreau envelop has the following properties:

**Lemma 4.** *Let $\ell : \mathcal{M} \mapsto \mathbb{R}$ be a convex function and $G$-Lipschitz. Then the following hold: a) $\ell_\beta$ is convex, $2G$-Lipschitz and $\beta$-smooth. b)$\ell'_\beta(x) = \beta[x - \text{prox}_\ell^\beta(x)]$. c) For all $x \in \mathcal{M}$, $\ell_\beta(x) \leq \ell(x) \leq \ell_\beta(x) + \frac{G^2}{2\beta}$.*

Note that the previous Moreau envelope is for one-dimensional functions while our surrogate loss $\ell$ is $d$ dimensional. Thus, for fixed $(x, y)$, we denote $g^y(\langle x, w \rangle) = \ell(w; x, y)$ in (2) and we calculate the Moreau envelop of $g^y(\cdot)$ instead, which is denoted as $g_\beta^y(\cdot)$. By Lemma 4 and since $\ell$ is Lipschitz and convex, we have $g_\beta^y$ is $2B$-Lipschitz and $\beta$-smooth. Thus, we have the following facts.

**Lemma 5.** *For any fixed $(x, y)$, denote $\ell(w; x, y) = g^y(\langle x, w \rangle)$ and $g_\beta^y(\cdot)$ as the Moreau envelop of $g^y(\cdot)$ with parameter $\beta$ and $\mathcal{M} = \mathbb{R}$. Let $f_\beta(w; x, y) = g_\beta^y(\langle w, x \rangle)$, then we have $f_\beta$ is $2B$-Lipschitz, $\beta$-smooth and $|f_\beta(w; x, y) - \ell(w; x, y)| \leq \frac{2B^2}{\beta}$ for all $w \in \mathbb{R}^d$.*

---

**Algorithm 4** $\mathcal{O}_{\beta,\gamma}$ : Gradient Oracle for $f_\beta(w; x, y)$

---

**Require:** Parameter vector $w \in \mathbb{R}^d$, data sample $(x, y)$ associate with $g_\beta^y(\cdot)$.

1: Let $m = \langle w, x \rangle$ and $\mathcal{Q} = [m - \frac{4B}{\beta}, m + \frac{4B}{\beta}]$.

2: Let $T = \frac{144B^2}{\gamma^2}$.

3: **for** $t = 1, 2, \cdots, T$ **do**

4:     $y_{t+1} = w_t - \eta_t(\sigma(w_t) - y + \beta(w_t - m))$ where $\eta_t = \frac{2}{\beta(t+1)}$.

5:     $w_{t+1} = y_{t+1}$ if $y_{t+1} \in \mathcal{Q}$, $w_{t+1} = m - \frac{4B}{\beta}$ if $y_{t+1} < m - \frac{4B}{\beta}$ and $w_{t+1} = m + \frac{4B}{\beta}$ otherwise.

6: **end for**

7: Denote $\bar{w} = \sum_{t=1}^{T} \frac{2t}{T(T+1)} x_t$.

8: Return $x\beta[m - \bar{w}]$.

---

One possible approach based on Lemma 5 is to obtain a smooth loss function $f_\beta(w; x, y)$, and then use Algorithm 2 and Algorithm 3 to obtain private estimators that achieve a small excess population risk for $f_\beta(w; x, y)$, i.e., $\mathbb{E}[f_\beta(w; x, y)] - \min_{w \in \mathbb{R}^d} \mathbb{E}[f_\beta(w; x, y)]$. However, there is a challenge: To use Algorithm 2, we need to calculate the gradient of $f_\beta(w; x, y)$, which is inefficient as it is hard to compute the proximal operator explicitly by Lemma 4. In the convex GLM case, Bassily et al. (2021) used the bisection method to calculate $\nabla f_\beta(w; x, y)$. However, this approach cannot be used here as it requires access to the function $g^y(\cdot)$, which involves integration for our problem and is difficult to compute accurately. In the following, we propose an algorithm that can efficiently approximate $\nabla f_\beta(w; x, y)$.

The idea is that by our definition we have $\nabla f_\beta(w; x, y) = x g_\beta'^y(\langle w, x \rangle)$, where $g_\beta'^y(m) = \beta[m - \text{prox}_g^\beta(m)]$. Thus, it is sufficient to approximate $\text{prox}_\ell^\beta(x)$ for given $x$. Recall that by the definition $\text{prox}_g^\beta(m) = \arg\min_{u \in \mathbb{R}}[g^y(u) + \frac{\beta}{2}|u - m|^2]$. We can show that $\text{prox}_g^\beta(m) \in \mathcal{Q} = [m - \frac{4B}{\beta}, m + \frac{4B}{\beta}]$ which indicates that

$$\text{prox}_g^\beta(m) = \arg\min_{u \in \mathcal{Q}}[g^y(u) + \frac{\beta}{2}|u - m|^2]$$

Thus, we can use the projected gradient descent (PGD) to solve the above strongly convex objective function. See Algorithm 4 for details. Based on the convergence rate of PGD, we have the following lemma:

**Lemma 6.** *Given any $\beta, \gamma > 0$. Then the gradient oracle $\mathcal{O}_{\beta,\gamma}$ for $f_\beta(w; x, y)$ in Algorithm 4 satisfies that $\|\nabla f_\beta(w; x, y) - \mathcal{O}_{\beta,\gamma}(w; x, y)\|_2 \leq \gamma$ for any fixed $w, x, y$. Moreover, $\mathcal{O}_{\beta,\gamma}$ has running time $O(d\frac{B^2}{\gamma^2})$.*

---

**Algorithm 5** Phased SGD for general non-convex GLM

---

**Require:** Private dataset: $D = \{(x_i, y_i)\}_{i=1}^n$, link function $\sigma$ satisfies Assumption 1 and is differentiable; privacy parameters $0 < \epsilon, \delta \leq 1$.

1: Set $k = \lceil \log_2(n) \rceil$, partite the whole dataset S into k subsets $\{D_1, \cdots, D_k\}$. Denote $n_i$ as the number of samples in $D_i$, i.e., $|D_i| = n_i$ where $n_i = \lfloor 2^{-i}n \rfloor$. Take a random initial vector $w_0 \in \mathcal{W}$.

2: **for** $i = 1, \cdots, k$ **do**

3:     Let $\eta_i = \frac{\eta}{4^i}$ and $w_i^1 = w_{i-1}$.

4:     **for** $t = 1, \cdots, n_i$ **do**

5:         Recall the oracle in Algorithm 4 for $f_\beta(w_i^t; x_i^t, y_i^t)$ in Lemma 5 with error $\gamma$ and denote it as $\tilde{\nabla} f_\beta(w_i^t; x_i^t, y_i^t)$. Update $w_i^{t+1} = w_i^t - \eta_i \tilde{\nabla} f_\beta(w_i^t; x_i^t, y_i^t)$, where $(x_i^t, y_i^t)$ is the $t$-th sample of $D_i$.

6:     **end for**

7:     Denote $w_i = \bar{w}_i + \zeta_i$, where $\bar{w}_i = \frac{1}{n_i} \sum_{t=1}^{n_i} w_i^t$ and $\zeta_i \sim \mathcal{N}(0, \tau_i^2 \mathbb{I}_d)$ with $\tau_i = \frac{10 B R \eta_i \sqrt{\log \frac{1}{\delta}}}{\epsilon}$.

8: **end for**

---

Using the previous Lemma 6, one possible approach is to use the approximate oracle in Algorithm 2, which is the main idea behind Algorithm 5. However, there is another issue: if we use the same proof as in the previous case where the link function has a subgradient everywhere, we can only obtain an upper bound that depends on $\|w_\beta^*\|_2$, where $w_\beta^* = \arg\min_{w \in \mathbb{R}^d} \mathbb{E}[f_\beta(w; x, y)]$. This phenomenon has also been observed in GLMs with convex and non-smooth loss functions Bassily et al. (2021). Fortunately, we can conduct a

---

**Algorithm 6** DP-Projected Phased SGD for general non-convex GLM

---

**Require:** Private dataset: $D = \{(x_i, y_i)\}_{i=1}^n$, link function $\sigma$ satisfies Assumption 1 and is differentiable; privacy parameters $0 < \epsilon, \delta \leq 1$.
  1: Sample a fast JL matrix $\Phi \in \mathbb{R}^{m \times d}$, and denote $\tilde{D} = \{(\Phi x_1, y_1), \cdots, (\Phi x_n, y_n)\}$.
  2: Run Algorithm 5 on the projected dataset $\tilde{D}$, where in Line 8, $\zeta_i \sim \mathcal{N}(0, \tau_i^2 \mathbb{I}_m)$ with $\tau_i = \frac{20B\eta_i \sqrt{\log \frac{2}{\delta}}}{\epsilon}$ and replace $B$ by $2B$ in Algorithm 4. Denote the output as $w_k$.
  3:
  4: Return $\hat{w} = \Phi^T w_k$

---

finer analysis of the theoretical guarantee of Algorithm 2 and show that we can obtain an upper bound that depends on $W$ instead of $\|w_\beta^*\|_2$. Similar to the above results, we have the following two results.

**Theorem 4.** *Under Assumption 1, for any $0 < \epsilon, \delta \leq 1$, Algorithm 5 is $(\epsilon, \delta)$-DP. Moreover, when $\eta = O(\min\{\frac{\epsilon}{\sqrt{\theta \log \frac{1}{\delta}}}, \frac{1}{\sqrt{n}}\}) \leq \frac{2}{\beta}$, $\gamma = O(\frac{1}{n \log n})$ and $\beta = O(\sqrt{n})$. Then we have*

$$\mathbb{E}L_\mathcal{P}(w_k) - L_\mathcal{P}(w^*) \leq O\Big(\frac{\sqrt{\theta \log \frac{1}{\delta}}}{n\epsilon} + \frac{1}{\sqrt{n}}\Big). \tag{3}$$

**Theorem 5.** *Under Assumption 1 and let $m = O(\log(\frac{n}{\delta})(n\epsilon)^{\frac{2}{3}})$, Algorithm 6 is $(\epsilon, \delta)$-DP for any $0 < \epsilon, \delta \leq 1$. Moreover, when $\eta = O(\min\{\frac{\epsilon}{\sqrt{m \log \frac{1}{\delta}}}, \frac{1}{\sqrt{n}}\}) \leq \frac{1}{\beta}$, $\gamma = O(n \log n)$ and $\beta = O(\sqrt{n})$ then we have*

$$\mathbb{E}L_\mathcal{P}(w_k) - L_\mathcal{P}(w^*) \leq \tilde{O}\Big(\frac{1}{\sqrt{n}} + \frac{\sqrt{\log \frac{1}{\delta}}}{(n\epsilon)^{\frac{2}{3}}}\Big).$$

### 4.1.3 ReLU Link Function

In the previous sections, we focused on the case where the link function in model (1) satisfies Assumption 1. Although this assumption includes several commonly used activation functions, it excludes the ReLU function where $\sigma(x) = \max\{0, x\}$ due to the boundedness assumption of $\sigma$. Here, we will consider the ReLU link function since it is a standard activation function in neural networks.

Similar to Assumption 1, we still assume that $\|w^*\|_2 \leq W$, $\|x\|_2 \leq 1$, and $y \in [-B, B]$. Note that since ReLU is Lipschitz, we can still use Lemma 3, and it is sufficient to consider the problem of minimizing $L_\mathcal{P}^\ell(w)$. However, the main difficulty now is that the surrogate loss function $\ell(w; x, y)$ is no longer Lipschitz over the whole space $\mathbb{R}^d$, as $\nabla \ell(w; x, y) = (\sigma(\langle w, x \rangle) - y)x$ is unbounded. Thus, our above methods cannot be used for the ReLU case, as all of them need to assume that $\ell(w; x, y)$ is Lipschitz over the whole space. This is due to the fact that $\bar{w}$ and $w_i^t$ in Algorithm 2 may not lie in the constraint set $\mathcal{W}$.

To address the issue, we make the key observation that although the surrogate loss function $\ell(w; x, y)$ is not Lipschitz over the whole space, it will be Lipschitz over bounded sets. Specifically, for any $w$ with $\|w\|_2 \leq W$, we have $\|\nabla \ell(w; x, y)\|_2 \leq W + B$. Based on this observation, we propose to constrain $w$ over a bounded domain during updates. To achieve this, we adopt the DP version of projected gradient descent (DP-PGD) introduced in Bassily et al. (2014), which adds noise to the gradient and performs the projection operation after updating the model, thereby enforcing $w$ to be bounded during each iteration. Building on Algorithm 3, we preprocess the data with a JL matrix and project all feature vectors onto an $m$-dimensional space before applying DP-PGD. Finally, we lift the private estimator to the original space after the DP-PGD algorithm. See Algorithm 7 for the full details.

**Theorem 6.** *Algorithm 7 is $(\epsilon, \delta)$-DP for any $0 < \epsilon, \delta \leq 1$ under the previous assumptions and $m = O(\log(\frac{n}{\delta})(n\epsilon)^{\frac{2}{3}})$. Moreover, take $\eta = O(\frac{1}{\sqrt{T}}) \leq \frac{1}{2}$ and $T = O(\min\{n, \frac{n^2\epsilon^2}{m \log 1/\delta}\})$, in Algorithm 7 we have*

$$\mathbb{E}L_\mathcal{P}(\bar{w}) - L_\mathcal{P}(w^*) \leq \tilde{O}\Big(\frac{1}{\sqrt{n}} + \frac{1\sqrt{\log 1/\delta}}{(n\epsilon)^{\frac{2}{3}}}\Big).$$

---

**Algorithm 7** DP-Projected GD for ReLU Regression

---

**Require:** Private dataset: $D = \{(x_i, y_i)\}_{i=1}^n$, ReLU link function $\sigma(w) = \max\{0, w\}$; privacy parameters $0 < \epsilon, \delta \leq 1$.

1: Sample a JL matrix $\Phi \in \mathbb{R}^{m \times d}$, and denote $\tilde{D} = \{(\Phi x_1, y_1), \cdots, (\Phi x_n, y_n)\}$.
2: **for** $t = 1, \cdots, T$ **do**
3:      Update $\tilde{w}_{t+1}$ as $\tilde{w}_{t+1} = \Pi_{\tilde{W}}[\tilde{w}_t - \eta(\frac{1}{n}\sum_{i=1}^n(\max\{0, \langle \tilde{w}_t, \Phi x_i \rangle\} - y_i)\Phi x_i + \zeta_t)]$, where $\zeta \sim \mathcal{N}(0, \sigma^2 \mathbb{I}_m)$
     with $\sigma^2 = \frac{32(4W+B)^2 T \log \frac{2}{\delta}}{n^2 \epsilon^2}$, $\tilde{W} = \{w \in \mathbb{R}^m | \|w\|_2 \leq 2W\}$ and $\Pi$ is the projection operator.
4: **end for**
5:
6: Return $\bar{w} = \frac{\sum_{t=1}^T \Phi^T \tilde{w}_T}{T}$.

---

By combining the error bound of DP-PGD in Bassily et al. (2014) with our analysis, we obtain a bound of $\tilde{O}(n^{-\frac{1}{2}} + \min\{\sqrt{d}(n\epsilon)^{-1}, (n\epsilon)^{-\frac{2}{3}}\})$. This bound is worse than those derived in the previous section because the ReLU link function is not Lipschitz over $\mathbb{R}^d$. It is worth noting that Arora et al. (2022b) also employs DP-PGD for convex generalized linear loss and obtains the same bound. However, their analysis assumes the loss function to be non-negative, whereas our loss function $\ell$ in (2) does not satisfy this assumption.

## 4.2 Misspecified Model

In previous sections, we focused on model (1) where $\mathbb{E}[y|x] = \sigma(\langle w^*, x \rangle)$. However, such an assumption is quite strong. Instead of the well-specified model, we always encounter the misspecified one that does not directly impose any probability condition on the label generating process. [4] Since the zero-mean random noise assumption does not hold, we cannot apply Lemma 3, which transforms the original population risk to the population risk of a convex surrogate loss. As a result, none of the above methods can be used in this case. A natural question is, *what are the theoretical behaviors of GLMs in the misspecified model?*

In fact, even in the non-private case, the problem of GLMs in the misspecified model is quite challenging and is still not well understood in general. Thus, rather than considering upper bounds for general loss functions, we aim to illustrate the differences with the well-specified model by examining specific losses. In particular, we will study ReLU regression in the misspecified model.

Similar to the previous section, we will still examine the squared population risk function $L_{\mathcal{P}}(w) = \mathbb{E}_{(x,y) \sim \mathcal{P}}(\sigma(\langle w, x \rangle) - y)^2$. It is noteworthy that Manurangsi & Reichman (2018) shows that in the absence of distributional assumptions on the marginal distribution of $x$, i.e., $\mathcal{P}_x$, finding a parameter $w$ such that $L_{\mathcal{P}}(w) \leq O(L_{\mathcal{P}}(w^*)) + \alpha$ with some small error $\alpha$ is NP-hard even in the non-private case. Therefore, compared to the well-specified model, we need additional assumptions on $\mathcal{P}_x$, and we will concentrate on the following isotropic log-concave distributions, which include uniform distribution over $[0, 1]^d$ and Bernoulli distribution.

**Assumption 2.** We assume the marginal distribution of $x$ is isotropic log-concave, i.e., $\mathbb{E}_{\mathcal{P}_x}[x] = 0$ and $\mathbb{E}_{\mathcal{P}_x}[xx^T] = \mathbb{I}_d$, and its density function $f$ satisfies $f(\lambda x + (1-\lambda)y) \geq f(x)^\lambda f(y)^{1-\lambda}$ for every $x, y \in \text{Supp}(\mathcal{P}_x)$ and $\lambda \in [0, 1]$. Moreover, we assume $\|x\|_2 \leq \sqrt{d}$ and $y \in [-B, B]$.

To illustrate our idea, we first provide some notations. For any function $f : \mathbb{R}^d \mapsto \mathbb{R}$ and distribution $\mathcal{P}$ for $(x, y)$, we denote $\chi_{\mathcal{P}}^f = \mathbb{E}_{x \sim \mathcal{P}_{\mathcal{X}}}[f(x)x]$, $\chi_{\mathcal{P}}^{\sigma_w} = \mathbb{E}_{x \sim \mathcal{P}_{\mathcal{X}}}[\sigma(\langle w, x \rangle)x]$ and $\chi_{\mathcal{P}} = \mathbb{E}_{\mathcal{P}}[yx]$. Our method is motivated by the following observations. Firstly, we can show that if $\mathcal{P}_{\mathcal{X}}$ is isotropic, then for any vector $w$ the distance between $\chi_{\mathcal{P}}^{\sigma_w}$ and $\chi_{\mathcal{P}}$ is bounded by $\sqrt{L_{\mathcal{P}}(w)}$, i.e., $\|\chi_{\mathcal{P}}^{\sigma_w} - \chi_{\mathcal{P}}\|_2 \leq \sqrt{L_{\mathcal{P}}(w)}$. Secondly, for ReLU regression, there exists a constant $\mu > 0$ such that $\sigma$ is $\mu$-strongly convex w.r.t $\mathcal{P}_{\mathcal{X}}$ if the marginal distribution is isotopic and log-concave Diakonikolas et al. (2020), i.e., for any $w, v$ we have $\langle \chi_{\mathcal{P}}^{\sigma_w} - \chi_{\mathcal{P}}^{\sigma_v}, w - v \rangle \geq \mu \|w - v\|_2^2$.

---

[4]This setting is also known as the agnostic setting in literature Diakonikolas et al. (2020); Goel et al. (2019).

Under Assumption 2 and the above strong convexity we can show that for any vector $w$,

$$\mathbb{E}_{\mathcal{P}}[(\sigma(\langle w, x\rangle) - \sigma(\langle w^*, x\rangle))^2] \leq O(\|\chi_{\mathcal{P}}^{\sigma_w} - \chi_{\mathcal{P}}^{\sigma_{w^*}}\|_2^2)$$
$$\leq O(\|\chi_{\mathcal{P}}^{\sigma_w} - \chi_{\mathcal{P}}\|_2^2 + \|\chi_{\mathcal{P}}^{\sigma_{w^*}} - \chi_{\mathcal{P}}\|_2^2)$$
$$= O(\|\chi_{\mathcal{P}}^{\sigma_w} - \chi_{\mathcal{P}}\|_2^2 + L_{\mathcal{P}}(w^*)).$$

Thirdly, by the triangle inequality we can easily find that

$$L_{\mathcal{P}}(w) \leq O(L_{\mathcal{P}}(w^*) + \mathbb{E}_{\mathcal{P}}[(\sigma(\langle w, x\rangle) - \sigma(\langle w^*, x\rangle))^2]).$$

Thus, in total we have for any vector $w$,

$$L_{\mathcal{P}}(w) \leq O(L_{\mathcal{P}}(w^*) + \|\chi^{\sigma_w} - \chi_{\mathcal{P}}\|_2^2).$$

Moreover, we can easily get that $\chi^{\sigma_w} - \chi_{\mathcal{P}}$ is the gradient of the population risk function of the surrogate loss function in (2), i.e., $\nabla L_{\mathcal{P}}^{\ell}(w) = \chi^{\sigma_w} - \chi_{\mathcal{P}}$. Thus, it is sufficient for us to find a private estimator $w$ to make $\|\nabla L_{\mathcal{P}}^{\ell}(w)\|_2$ be as small as possible.

Although some previous studies have addressed finding a first-order stationary point privately for population risk functions, such as Wang & Xu (2019); Kang et al. (2021); Arora et al. (2022a), their methods cannot be applied to our function $L_{\mathcal{P}}^{\ell}$ because they assume that the loss is Lipschitz over $\mathbb{R}^d$, which is not the case for our loss. To overcome this challenge, we present a new algorithm, Adaptive DP Batched Gradient Descent (Algorithm 8). The main idea is to partition the dataset into several subsets and, in each iteration, use one subset for private Gradient Descent. Although our loss function $\ell(w; x, y)$ is not uniformly Lipschitz over $\mathbb{R}^d$, we can still find that $\|\nabla \ell(w_{t-1}; x, y)\|_2 \leq \sqrt{d}\|w_{t-1}\|_2 + B$, whose upper bound only depends on the current model $w_{t-1}$. Therefore, we can still use the Gaussian mechanism with sensitivity $2(\|w_{t-1}\|_2 + B)$ to the gradient to ensure $(\epsilon, \delta)$-DP. Our algorithm is fundamentally different from previous DP-GD based methods Bassily et al. (2014); Wang et al. (2017), as the Gaussian noise added also depends on the current model. In general, our method can provide a tighter bound, as $\|w_{t-1}\|_2$ becomes smaller as $t$ increases, which implies that we add smaller noise to the gradient.

---

**Algorithm 8** Adaptive DP Batched Gradient Descent

**Require:** Private dataset: $D = \{(x_i, y_i)\}_{i=1}^n$, ReLU link function $\sigma$; privacy parameters $0 < \epsilon, \delta \leq 1$.
 1: Partite the data $D$ into $T$ subsets $\{D_1, \cdots, D_T\}$ where $m = |D_i| = \frac{n}{T}$.
 2: Denote the loss function $\ell$ as $\ell(w; x, y) = \int_0^{\langle w, x\rangle}(\sigma(z) - y)dz$. Initialize $w_0 = 0$.
 3: **for** $i = 1, \cdots, T$ **do**
 4:     Let $w_i = w_{i-1} - \eta(\nabla L^{\ell}(w_{i-1}; D_i) + \zeta_{i-1}) = w_{i-1} - \eta(\frac{1}{m}\sum_{x \in D_i}(\max\{0, \langle w_{i-1}, x_i\rangle\} - y_i)x_i + \zeta_{i-1})$,
     where $\zeta_{i-1} \sim \mathcal{N}(0, \sigma_{i-1}^2 I_d)$ with $\sigma_{i-1} = \frac{8(\sqrt{d}\|w_{t-1}\|_2 + B)^2 \log(1.25/\delta)}{m^2\epsilon^2}$.
 5: **end for**
 6:
 7: Return $w_T$

---

Combining with all the above ideas, we can show the following result for Algorithm 8.

**Theorem 7.** *Consider ReLU regression and assume Assumption 2 holds. For any $0 < \epsilon, \delta < 1$, Algorithm 8 is $(\epsilon, \delta)$-DP. Moreover denote $w_{\ell}^* = \arg\min_{w \in \mathbb{R}^d} L_{\mathcal{P}}^{\ell}(w)$ with $\ell$ in (2). For any error $\alpha \in (0, \|w_{\ell}^*\|_2)$, if $n$ is sufficiently large such that*

$$n \geq \tilde{\Omega}(\max\{\frac{d\|w_{\ell}^*\|_2\sqrt{\log \frac{1}{\delta} \log \frac{1}{\zeta}}}{\epsilon\alpha}, \frac{\|w_{\ell}^*\|_2^2 d \log^4 \frac{1}{\zeta}}{\alpha^2}\}),$$

*setting $T = O(\log(\|w_{\ell}^*\|_2))$ and $\eta \leq \frac{1}{16}$ in Algorithm 8 we have*

$$L_{\mathcal{P}}(w_T) \leq 2(1 + 2\mu)L_{\mathcal{P}}(w^*) + \alpha$$

*with probability at least $1 - \zeta$ with $\zeta \geq \exp(-O(\sqrt{d}))$.*

In Theorem 7, we demonstrate that for ReLU regression under Assumption 2, the sample complexity required to achieve $L_{\mathcal{P}}(w) - c \cdot L_{\mathcal{P}}(w^*) \leq \alpha$ with some $c > 0$ is $\tilde{O}(\max\{\frac{d}{\epsilon\alpha}, \frac{d}{\alpha^2}\})$. There are several differences in comparison to the results in previous sections. Firstly, here we can only obtain a bound for $L_{\mathcal{P}}(w) - O(L_{\mathcal{P}}(w^*))$ instead of the original excess population risk. In fact, this big-$O$ term is necessary, as Goel et al. (2019) provides hardness results for $L_{\mathcal{P}}(w) - L_{\mathcal{P}}(w^*) \leq \alpha$ with $\alpha \in (0,1)$, even if the underlying distribution is the standard Gaussian. The second difference is that unlike the previous results, where sample complexities are independent of $d$, the sample complexity here depends linearly on $d$. This dependency results from two factors: the magnitude of noise added depends on $\sqrt{d}$, and the estimation error of $\|\nabla L^\ell(w; D)\|_2$ introduces an additional $d$ factor. We cannot use the same strategy as in Algorithm 2 since projecting the data will alter the sample distribution and destroy the strongly convex property. Therefore, even in the non-private case, there is still a factor of $d$ in the sample complexity.

## 5 Extension to Two-layer Neural Networks

In this section, we present an extension of our previous methods to one-hidden layer fully connected neural networks. Our focus is mainly on the cases where the activation functions are either sigmoid or ReLU. We restrict ourselves to the well-specified model. Before presenting the details, we start by extending our model (1) to a bounded noise setting in a high-dimensional feature space. We assume $\mathcal{K}$ is a kernel function in a Reproducing Kernel Hilbert Space (RKHS) $\mathcal{H} \subseteq \mathbb{R}^k$ with some $k$, and $\psi(\cdot) \in \mathbb{R}^k$ is the corresponding feature map satisfying $|\psi(x)|_2 \leq 1$ for all $x \in \mathcal{P}_{\mathcal{X}}$. We consider the following model:

$$y = \sigma(\langle w^*, \psi(x)\rangle + \phi(x)) + \zeta, \tag{4}$$

where $w^* \in \mathcal{H}$ is the underlying parameter with $\|w^*\|_2 \leq W$, $\phi(x)$ is a noise function which satisfies $|\phi(x)| \leq M$, $\zeta$ is a random noise whose mean is 0 and $\sigma$ is a (non-convex) link function. Note that in the case of $\phi(x) = 0$ and $\psi$ is the identity function, (4) is equivalent to model (1). Similar to the previous section, here we consider the squared loss where $L_{\mathcal{P}}(h) = \mathbb{E}_{(x,y)\sim\mathcal{P}}[(h(x) - y)^2]$ for any function $h$ [5] and we want to minimize the excess population risk:

$$L_{\mathcal{P}}(h) - \min_h L_{\mathcal{P}}(h) = \mathbb{E}_{(x,y)\sim\mathcal{P}}[(h(x) - \sigma(\langle w^*, \psi(x)\rangle + \phi(x)))^2].$$

We consider the model (4) because, as we will show later, for some one-hidden layer neural networks, we can always find $w^*$, $\psi(\cdot)$, and $M$ to approximate the hidden layer, and the link function $\sigma$ can be viewed as the activation function of the output layer. We first present the following assumption for this section.

**Assumption 3.** We assume that there exist constants $W, G = O(1)$ such that $\|w^*\|_2 \leq W$, $y \in [0,1]$ [6] and the link function $\sigma : \mathbb{R} \mapsto [0,1]$ is $G$-Lipschitz and non-monotone decreasing, and has sub-gradient everywhere. Moreover, in model (4) we assume $\|\psi(x)\|_2 \leq 1$ and $\|\phi(x)\|_2 \leq M$ for every $x \sim \mathcal{P}_x$.

To minimize the population risk, similar to the previous section, we consider the surrogate loss

$$\ell(w; x, y) = \int_0^{\langle w, \psi(x)\rangle} (\sigma(z) - y)dz. \tag{5}$$

By Lemma 2 we can see the $\ell$ is 1-Lipschitz and $G$-smooth. Similar to Lemma 3, the following lemma shows the relation between the original population risk and the population risk for the surrogate loss.

**Lemma 7.** *For any $w \in \mathcal{H}$ we have*

$$L_{\mathcal{P}}(w) - \min_h L_{\mathcal{P}}(h) \leq 4G(L_{\mathcal{P}}^\ell(w) - L_{\mathcal{P}}^\ell(w^*)) + 2G^2M^2 + 4GM.$$

By Lemma 7, we can see that it is sufficient to find a private model that minimizes the difference between $L_{\mathcal{P}}^\ell(w)$ and $L_{\mathcal{P}}^\ell(w^*)$. To achieve this goal, we can use a similar algorithm as presented in Algorithm 1, with the main difference being the use of $\tilde{D} = (\psi(x_i), y_i)_{i=1}^n$ instead of the raw data. For more details, please refer to Algorithm 9. Similar to Theorem 1 and 2 we have the following result.

---

[5]Note that since we need to estimate both $w^*$ and $\phi$, here we use a function instead of vector in the previous section.

[6]Note that here we assume $y$ and $\sigma$ is in $[0,1]$ is that there are commonly used in practice. We can extend to any bounded interval.

---

**Algorithm 9** DP Two-layer Neural Networks

---

**Require:** Private dataset: $D = \{(x_i, y_i)\}_{i=1}^n$, link function $\sigma$ satisfies Assumption 3; privacy parameters $0 < \epsilon, \delta \leq 1$, $\theta$ is an upper bound of the expected rank $\sum_{i=1}^n \psi(x_i)\psi(x_i)^T$.
  1: Denote the data $\tilde{D} = \{(\psi(x_i), y_i)\}_{i=1}^n$. If $\epsilon > \frac{\theta}{n}$, run Algorithm 2 with $\tilde{D}$. Otherwise run Algorithm 3 with $\tilde{D}$.

---

**Theorem 8.** *Under Assumption 3, for any $0 < \epsilon, \delta \leq 1$, Algorithm 9 is $(\epsilon, \delta)$-DP. Moreover we have its output $w$ satisfies*

$$\mathbb{E}L_{\mathcal{P}}(w) - \min_h L_{\mathcal{P}}(h) \leq \tilde{O}(\min\{\frac{\sqrt{\theta \log \frac{1}{\delta}}}{n\epsilon}, \frac{\sqrt{\log \frac{1}{\delta}}}{\sqrt{n\epsilon}}\}$$

$$+ \frac{1}{\sqrt{n}} + M^2 + M),$$

*where $\theta$ is an upper bound on the expected rank of $\sum_{i=1}^n \psi(x_i)\psi(x_i)^T$ if $\eta = O(\min\{\frac{\epsilon}{\sqrt{\theta \log \frac{1}{\delta}}}, \frac{1}{\sqrt{n}}\}) \leq \frac{1}{G}$ in Algorithm 2 and $\eta = O(\min\{\frac{\epsilon}{\sqrt{m \log \frac{1}{\delta}}}, \frac{1}{\sqrt{n}}\}) \leq \frac{1}{G}$ in Algorithm 3 with $m = O(\log(n/\delta)n\epsilon)$.*

*Remark* 9. It is worth noting that the rate of sample size $n$ in Theorem 8 is lower than that in Theorem 1 ($n^{-\frac{1}{2}}$ v.s. $n^{-\frac{2}{3}}$). This is due to that in the noiseless case ($M = 0$), $w^*$ is also a global minimizer of $L_{\mathcal{P}}^\ell(w)$, which is not the case in model (4). Therefore, we cannot rely on this property and the smooth Lipschitz condition to demonstrate that the error caused by projecting onto a lower space is $\tilde{O}(\frac{1}{m})$. Instead, we can only use the Lipschitz condition to obtain an error of $\tilde{O}(\frac{1}{\sqrt{m}})$.

One question is as we know $L_{\mathcal{P}}^\ell(w) - L_{\mathcal{P}}^\ell(w^*) \leq L_{\mathcal{P}}^\ell(w) - \min_{w \in \mathbb{R}^k} L_{\mathcal{P}}^\ell(w)$, why we do not consider to bound the latter term? Actually, considering the latter term will make us get an error that depends on $\|w_\ell^*\|_2$ with $w_\ell^* = \arg\min_{w \in \mathbb{R}^k} L_{\mathcal{P}}^\ell(w)$, whose upper bound is unknown. Thus, we need to analyze $L_{\mathcal{P}}^\ell(w) - L_{\mathcal{P}}^\ell(w^*)$ directly. Fortunately, by giving a finer analysis for the Phased SGD we can get such an upper bound.

Assuming that the term $\phi(x)$ is a noise function that is bounded by a sufficiently small constant $M$, Theorem 8 implies that if a function $f$ can be approximated by an element of an appropriate RKHS, then Algorithm 8 can be used to obtain a private estimator. This is formalized in the following corollary.

**Definition 7** (($M, W$)-Uniform Approximation). Let $f$ be a function mapping from domain $\mathcal{X}$ to $\mathbb{R}$ and $\mathcal{P}_{\mathcal{X}}$ be a distribution over $\mathcal{X}$. Let $\mathcal{K}$ be a kernel function with corresponding RKHS $\mathcal{H} \subseteq \mathbb{R}^k$ and feature vector $\psi$. We say $f$ is ($M, B$)-uniformly approximated by $\mathcal{K}$ over $\mathcal{P}_{\mathcal{X}}$ if there exists some $w^* \in \mathcal{H}$ with $\|w^*\|_2 \leq W$ such that for all $x \sim \mathcal{P}_{\mathcal{X}}$ we have $|f(x) - \langle w^*, \psi(x)\rangle| \leq M$.

**Corollary 1.** *Consider a distribution $\mathcal{P}$ such that $\mathbb{E}[y|x] = \sigma(f(x))$ where $\sigma$ is a known $G$-Lipshcitz and increasing function, and $f$ is ($M, W$)-approxiamted by some kernel function $\mathcal{K}$ and feature map $\psi$ such that $\mathcal{K}(x, x') \leq 1$. The function $h(x) = \sigma(\langle w, \psi(x)\rangle)$ for the output $w$ in Algorithm 9 achieves the same error bound as in Theorem 8.*

Next, we will apply Corollary 1 to some neural network models by using some recent results on approximation theory for neural networks Goel & Klivans (2019). We consider the following one-hidden layer neural networks with $k$-hidden units and one output node:

$$y = \mathcal{N}_2(x) + \zeta, \text{ where } \mathcal{N}_2 : x \mapsto \sigma_2(\sum_{t=1}^k b_t \sigma_1(\langle a_t, x\rangle)). \tag{6}$$

Here we assume $\|x\|_2 = 1$, $\|a_t\|_2 = 1$ for each $t$ and $\|b\|_2 = 1$ where $b = (b_1, \cdots, b_k)$, and $\sigma_1, \sigma_2$ are two activation functions. Here $\sigma_2$ satisfies the properties in Assumption 3 and $\sigma_1$ could be either Sigmod and ReLU activation functions. In the following, we provide sample complexities to achieve an error of $\alpha$ for these two cases.

**Theorem 10.** *Consider samples $\{(x_i, y_i)\}_{i=1}^n$ are i.i.d. drawn from distribution $\mathcal{P}$ such that $\mathbb{E}[y|x] = \mathcal{N}_2(x)$ with $\sigma_2 : \mathbb{R} \mapsto [0, 1]$ is a known $G$-Lipscitz and increasing function and $\sigma_1$ is the sigmoid function. Then*

*when* $n = O((\frac{kG}{\alpha})^{2C} \frac{\log 1/\delta}{\epsilon} + (\frac{Gk}{\alpha})^C \frac{\sqrt{\theta \log 1/\delta}}{\epsilon})$ *with some constant* $C > 0$ *we have*

$$\mathbb{E}L_{\mathcal{P}}(h) - \min_h L_{\mathcal{P}}(h) = \mathbb{E}L_{\mathcal{P}}(h) - L_{\mathcal{P}}(\mathcal{N}_2) \leq \alpha.$$

*Here* $h(x) = \sigma(\langle w, \psi(x) \rangle)$ *for some feature map* $\psi(x) \in R^{D_m}$ *with* $D_m = \tilde{O}(d^{O(\log \frac{k}{\alpha})})$ *and* $w$ *is the output of Algorithm 9 with the feature map* $\psi(\cdot)$.

**Theorem 11.** *Consider samples* $\{(x_i, y_i)\}_{i=1}^n$ *are i.i.d. drawn from distribution* $\mathcal{P}$ *such that* $\mathbb{E}[y|x] = \mathcal{N}_2(x)$ *with* $\sigma_2 : \mathbb{R} \mapsto [0, 1]$ *is a known* $G$-*Lipschitz and increasing function and* $\sigma_1$ *is the ReLU function. Then when* $n = O(4^{C(\frac{Gk}{\alpha})} \frac{\log 1/\delta}{\epsilon} + 2^{C(\frac{Gk}{\alpha})} \frac{\sqrt{\theta \log 1/\delta}}{\epsilon})$ *with some constant* $C > 0$ *we have*

$$\mathbb{E}L_{\mathcal{P}}(h) - \min_h L_{\mathcal{P}}(h) = \mathbb{E}L_{\mathcal{P}}(h) - L_{\mathcal{P}}(\mathcal{N}_2) \leq \alpha.$$

*Here* $h(x) = \sigma(\langle w, \psi(x) \rangle)$ *for some feature map* $\psi(x) \in R^{D_m}$ *with* $D_m = O(\frac{\sqrt{k}}{\alpha} d^{O(\frac{k}{\alpha})})$ *and* $w$ *is the output* $w$ *of Algorithm 9 with the feature map* $\psi(\cdot)$.

*Remark* 12. The results for one-hidden layer neural networks are quite intricate. Firstly, the sample complexity now depends on poly($k$) in the sigmoid case and depends on the exponential of $k$ and $\frac{1}{\alpha}$ in the ReLU case, which is due to the approximation errors using feature maps. However, it is noteworthy that, similar to the GLM case, the sample complexities are still independent of the data dimension. The second difference is that Algorithm 9 is inefficient in the ReLU case, as the dimension of the feature map will be exponential. Hence, developing efficient algorithms for privately learning one-hidden layer networks will remain an open problem.

## 6    DP-SGD for Multi-layer Neural Networks

In previous sections, we examined GLMs and one-hidden layer neural networks, but there are three critical issues with those results: (1) While we proposed several new algorithms, DP-SGD based methods Abadi et al. (2016) are preferred in practice for private neural network training. Can we obtain utility guarantees for vanilla DP-SGD in Abadi et al. (2016)? Alternatively, how do different factors such as the number of nodes, clipping threshold, and iteration number impact the utility theoretically? (2) Most of the aforementioned results rely on the well-specified model assumption and the squared loss in population risk, which can be too stringent in practice. Can we provide utility analysis without these assumptions? (3) Previous methods for one-hidden layer networks heavily depend on their specific forms and cannot be extended to general multi-layer structures. To address these issues, we study the utility of the projected version of DP-SGD for general multi-layer neural networks in this section.

We consider fully connected neural networks with depth (number of layers) $L$, width $m$ in each layer, and input data dimension $d$. Such a network could be represented by its weight matrices at each layer: For $L \geq 2$, let $\mathbf{W}_1 \in \mathbb{R}^{m \times d}$ be the weight matrix between the input layer and the first hidden layer, $\mathbf{W}_l \in \mathbb{R}^{m \times m}$ with $l = 2, \cdots, L-1$ as the weight matrices between hidden layers and $\mathbf{W}_L \in \mathbb{R}^{1 \times m}$ be the weight matrix between the last hidden layer to the output layer.[7] For simplicity we denote $\mathbf{W} = (\mathbf{W}_1, ..., \mathbf{W}_L)$. Then the neural network on sample $x$ can be written as

$$f(\mathbf{W}, x) = (\sqrt{m}) \cdot \mathbf{W}_L \sigma(\mathbf{W}_{L-1} \sigma(\mathbf{W}_{L-2}...\sigma(\mathbf{W}_1 x)...)),$$

where $\sigma(\cdot)$ is the entry-wise activation function. In this paper, for convenience, we only consider the ReLU activation function $\sigma(s) = \max\{0, s\}$, which is arguably one of the most difficult activation functions to analyze due to its non-smoothness. The general analysis framework is able to extend to other activation functions like tanh, and sigmoid, as long as the function is smooth almost everywhere.

Besides the neural network, we also have a non-negative, differentiable, and $S$-Lipschitz convex loss function $\ell(f(\mathbf{W}, x), y)$ (denoted as $\ell(\mathbf{W}; x, y)$) which measures the difference between the prediction of network and the ground truth. In total, now our excess population risk is defined as $\mathbb{E}_{(\mathbf{x}, y) \sim \mathcal{D}} \ell(\mathbf{W}; x, y) -$

---

[7]For simplicity, we assume the widths of each hidden layer are the same. Our result can be extended to the setting where the widths of each layer are not equal in the same order.

$\min_{\mathbf{W} \in \mathcal{W}} \mathbb{E}_{(\mathbf{x},y) \sim \mathcal{D}} \ell(\mathbf{W}; x, y)$. We consider the following assumption throughout the whole part, which is commonly used in the previous work on analyzing theoretical behaviors of multi-layer neural networks such as Chen et al. (2020); Cao & Gu (2019a).

**Assumption 4.** Assume $||x||_2 \leq 1$ for all $x \in \mathcal{P}_x$ and the parameter space of the network is $\mathcal{W} = \mathcal{B}(\mathbf{0}, R)$, i.e., for all $\mathbf{W} \in \mathcal{W} : ||\mathbf{W}_l||_F \leq R$, for all $l \in [L]$.

---

**Algorithm 10** DP-SGD for Multi-layer Neural Networks

---

**Require:** : Private dataset: $D$, convex set $\mathcal{W} = \mathcal{B}(\mathbf{0}, R)$. Parameters: learning rate $\eta$, mini-batch size $M$, iteration $T$, privacy parameter $\epsilon \leq 1$, $\delta \leq 1/n^2$, clipping constant $C$.

1: Generate each entry of $\mathbf{W}_l^{(0)}$ independently from $N(0, 2/m)$, $l \in [L-1]$. Generate each entry of $\mathbf{W}_L^{(0)}$ independently from $N(0, 1/m)$.

2: **for** $t = 0$ to $T - 1$ **do**

3:      For each data $(x_i, y_i) \in D$ sample it probability $p$. Denote the batch as $B_t$.

4:      For each $(x_j^{(t)}, y_j^{(t)}) \in B_t$, denote $g_t(\mathbf{x}_j^{(t)}) = \nabla \ell(\mathbf{W}^{(t)}; \mathbf{x}_j^{(t)}, y_j^{(t)})$.

5:      Let $\tilde{g}_t(x_j^{(t)}) = g_t(x_j^{(t)}) / \max(1, \frac{||g_t(x_j^{(t)})||_2}{C})$.

6:      Update weight matrices as $\mathbf{W}^{(\mathbf{t+1})} = \Pi_{\mathcal{W}}(\mathbf{W}^{(t)} - \eta \cdot (\frac{1}{|B_t|} \sum_{(x_j^{(t)}, y_j^{(t)}) \in B_t} \tilde{g}_t(x_j^{(t)}) + \mathbf{G}_t))$, where $\mathbf{G}_t \sim \mathcal{N}(\mathbf{0}, \sigma^2 \mathbb{I})$ drawn independently each iteration.

7: **end for**

8:

9: Return $\tilde{\mathbf{W}}_{priv} = \frac{1}{T} \sum_{t=1}^{T} \mathbf{W}^{(t)}$

---

We aim to provide an upper bound on the excess population risk for DP-SGD in Algorithm 10 instead of developing new algorithms. Note that there are slight differences between Algorithm 10 and the one in Abadi et al. (2016). First, in the first step of Algorithm 10, the initial weight matrices are i.i.d. sampled from a specific Gaussian distribution, which is crucial for our utility analysis. Secondly, in step 6, we need to perform the projection after using the noisy and clipped sub-sampled gradients to update our weight matrices. In fact, the projection step is also necessary for our analysis. Finally, instead of using the weight matrices in the last iteration, our output is the average of all the intermediate weight matrices. We use the average for convenience of analysis, but we can still obtain a similar utility for the last iteration weight matrices by using the same strategy as in Shamir & Zhang (2013).

The main idea of our utility analysis is based on recent developments in the Neural Tangent Kernel (NTK) technique Jacot et al. (2020), which explains the generalization behaviors and provides theoretical guarantees for SGD in overparameterized neural networks. To introduce the idea of NTK, we first recall the definition of a Neural Tangent Random Feature function.

**Definition 8** (Neural Tangent Random Feature)**.** Let $\mathbf{W}^{(0)}$ be generated via the initialization process in Algorithm 10. Then the Neural Tangent Random Feature (NTRF) function is defined as

$$f_{ntk}(\mathbf{W}, \boldsymbol{x}) = f(\mathbf{W}^{(0)}, \boldsymbol{x}) + \left\langle \partial_{\mathbf{W}} f(\mathbf{W}^{(0)}, \boldsymbol{x}), \mathbf{W} \right\rangle.$$

Consider the parameter space $\mathcal{B}(\mathbf{W}^{(0)}, \omega)$, the corresponding NTRF function class is denoted as

$$\mathcal{F}(\mathbf{W}^{(\mathbf{0})}, \omega) = \{f_{ntk}(\mathbf{W}, x) : \mathbf{W} \in \mathcal{B}(\mathbf{W}^{(0)}, \omega), ||x||_2 \leq 1\}.$$

Note that an NTRF function is linear. The idea of using NTK to analyze the generalization performance of overparameterized neural networks is based on the observation that the dynamic of wide neural networks under SGD is similar to that of the corresponding local linearization. In detail, let $W^{(t)}$ denotes the updated parameter vector after the $t$-th iteration, and $L_f = \sum_{(x,y) \in D} \ell(f(W^{(t)}; x), y)$ denotes the sum of loss with respect to $f$. Via continuous time gradient descent we have $W^{(t+\Delta t)} - W^{(t)} = -\eta \Delta t \frac{\partial L(t)}{\partial W}$. Since $\frac{\partial f(W^{(t)}, D_{\mathcal{X}})}{\partial t} = \nabla_W f(W^{(t)}, D_{\mathcal{X}}) \frac{\partial W^{(t)}}{\partial t}$, and by chain role $\frac{\partial W^{(t)}}{\partial t} = -\eta \nabla_W f(W^{(t)}, D_{\mathcal{X}})^T \nabla_{f(W^{(t)}, D_{\mathcal{X}})} L$, where

$$f(W^{(t)}, D_{\mathcal{X}}) = vec([f(W^{(t)}, x_i)]_{x \in [n]})$$

is the vector of $f(W^{(t)}, x_i)$. The evolution of the neural network $f$ and $f_{ntk}$ can be written by

$$\underbrace{\frac{\partial f(W^{(t)}, D_{\mathcal{X}})}{\partial t} = -\eta \Theta_t(D_{\mathcal{X}}, D_{\mathcal{X}}) \nabla_f L_{f(W^{(t)}, D_{\mathcal{X}})},}_{Gradient\ of\ Neural\ Network}$$

$$\underbrace{\frac{\partial f_{ntk}(W^{(t)}, D_{\mathcal{X}})}{\partial t} = -\eta \Theta_0(D_{\mathcal{X}}, D_{\mathcal{X}}) \nabla_f L_{f_{ntk}(W^{(t)}, D_{\mathcal{X}})},}_{Gradient\ of\ Local\ Linearization}$$

where

$$\Theta_0(\mathcal{X}, \mathcal{X}) = \mathbb{E}_{W^{(0)}} \nabla_W f(W^{(0)}, D_{\mathcal{X}}) \nabla_W f(W^{(0)}, D_{\mathcal{X}})^T$$

is the NTK matrix and $\Theta_t(D_{\mathcal{X}}, D_{\mathcal{X}}) = \nabla_W f(W^{(t)}, D_{\mathcal{X}}) \nabla_W f(W^{(t)}, D_{\mathcal{X}})^T$ is the empirical NTK. Recently, Arora et al. (2019) gives the first non-asymptotic convergence rate for the NTK matrix $\Theta_t$ and shows $||\Theta_t - \Theta_0||_F \to 0$ when $m$ is sufficiently large, *i.e.*, the empirical NTK is proved to converge to a deterministic kernel under the infinite width setting Jacot et al. (2020) with high probability. Based on this, when $m$ is sufficiently large, from the above two equations we can see a basic idea to approximate the gradients of neural networks is using their linearizations, which are convex. Moreover, we can also control the difference between $f(W^{(t)}, x)$ and $f_{ntk}(W^{(t)}, x)$ by the term $||\Theta_t - \Theta_0||_F$. Thus, via NTK, analyzing the utility of SGD for neural networks will become similar to analyzing the utility of SGD for convex loss. Motivated by the above intuition, we finally get the following theorem for the utility of Algorithm 10.

**Theorem 13.** *There exist constants $c_1, c_2$ so that given the number of steps $T$ and $q = M/n$, for any $\epsilon < c_1 q^2 T$, Algorithm 10 is $(\epsilon, \delta)$-DP for any $0 < \delta < 1$ if we have $\sigma_t \geq c_2 \frac{qC\sqrt{T \log(1/\delta)}}{|B_t|\epsilon}$. Moreover, for any $\xi \in (0, e^{-1}], 0 < \gamma_1, \gamma < 1$, and $R > 0$, there exists*

$$m^*(\xi, R, L, S, T, C) = \widetilde{\Omega}(Poly(S, L, R) T^7 C^{-8} [\log(1/\xi)]^3)$$

*such that if $C \leq O(\min\{SL\sqrt{m}, R\})$, $m \geq m^*$, $M \geq \Omega(\log \frac{T}{\gamma_1})$ and $n \geq \tilde{\Omega}(\frac{C(\sqrt{L}m + \sqrt{md})\sqrt{T \log(1/\gamma) \log(1/\delta)}}{R\epsilon})$, then with probability at least $1 - \xi - \gamma - \gamma_1$ over the randomness of the algorithm, the excess population risk $L_{\mathcal{D}}(\hat{\mathbf{W}}) - \min_{\mathbf{W} \in \mathcal{W}} L_{\mathcal{D}}(\mathbf{W})$ of the output in Algorithm 10 with step size $\eta = \Theta(\frac{\sqrt{L}R}{C\sqrt{mT}})$ is upper bounded by*

$$\underbrace{\sqrt{\frac{\log(\frac{1}{\xi})}{T}}}_{Convergence\ rate} + \underbrace{\inf_{f \in \mathcal{F}(\mathbf{W}^{(0)}, \frac{R}{\sqrt{m}})} \{\frac{1}{T} \sum_{i=1}^{T} \ell(f(\mathbf{x}_i), y_i)\}}_{Approximation\ error}$$

$$+ SL^{\frac{3}{2}} R \cdot \underbrace{\widetilde{O}(\frac{\max(L, \frac{d}{m}) \log(\frac{1}{\gamma}) \log(\frac{1}{\delta}) m^2 \sqrt{T}}{n^2 \epsilon^2})}_{Privacy\ error}. \tag{7}$$

*Remark* 14. Compared to the results in previous sections, Theorem 13 provides a more complex upper bound. This upper bound is composed of three terms: The first term represents the sum of convergence rate and sampling error. The second term is the minimum value of $\frac{1}{T} \sum_{i=1}^{T} \ell(f(\mathbf{x}_i), y_i)$ among all reference functions in the NTRF function class. This term arises due to the approximation error caused by using NTRF functions to approximate neural networks. The last term corresponds to the error resulting from the addition of extra noise to gradients to ensure differential privacy. It is notable that when $\epsilon = \infty$, i.e., when in the non-private case, our result will be $O(\inf_{f \in \mathcal{F}(\mathbf{W}^{(0)}, \frac{R}{\sqrt{m}})} \{\frac{1}{T} \sum_{i=1}^{T} \ell(f(\mathbf{x}_i), y_i)\} + \frac{1}{\sqrt{T}})$. Moreover, when $m, T \to \infty$, the error will tend to zero.

*Remark* 15. Compared to the convex case, the impact of parameters $T$ and $m$ on the bound in Theorem 13 is more complicated. For network width $m$, if it is large enough, then $f_{ntk}(\mathbf{W}^{(0)}, x)$ will converge to a well-trained neural network, as pointed out by Lee et al. (2020); Jacot et al. (2020); Lee et al. (2018). In the interpolation regime, the training error can be zero, which means the approximation error tends to zero as $m$ becomes sufficiently large. However, $m$ cannot be arbitrarily large because the privacy error depends on $\text{poly}(m)$. As for parameter $T$, it should not be too large or too small. When $T$ is large, the privacy

error increases, and when $T$ is small, the convergence error becomes large. Furthermore, the upper bound is independent of the clipping threshold $C$ because we assume that $C \leq R$ and the step size $\eta$ depends on $\frac{1}{C}$ (which implies that $C$ cannot be too small). Thus, when $C$ is in some range, it will not have a significant impact on performance. However, the effect of $C$, whether large or small, remains an open problem.

*Remark* 16. The main weakness of Theorem 13 is the assumption of $n \geq O(m)$, which contradicts the overparameterized setting in NTK theory, where the number of nodes could be far greater than the sample size $n$. To address this weakness, recent studies have proposed additional assumptions on the gradient or loss of neural networks, such as low-rank gradients Zhou et al. (2020) and restricted Lipschitz continuity Li et al. (2022). However, all of these works only analyze the excess population risk for convex loss functions. Since we can show, via NTK theory, that the loss function is locally convex and has a bounded gradient with large enough $m$ with high probability, we believe that it is possible to remove the dependency on the number of weights in the utility by combining our theoretical analysis with those assumptions. This will be left as future work.

### 6.1 Experimental Investigation

In order to validate the usefulness of the aforementioned theorem and investigate the effect of hyperparameters on the error, as mentioned in Remark 15, we conducted experiments using a three-layer MLP model on the MNIST dataset. The training set comprised 60,000 samples, with each sample represented by a 784-dimensional vector.

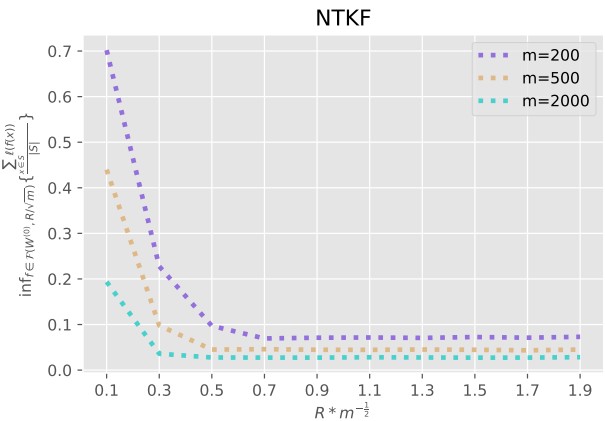

Figure 1: Results of the first term in Theorem 8.

1. First, we aim to study the NTRF approximation error in the bound of Theorem 13 with different values of $R$ and $m$, which can be approximated by solving the convex optimization problem $\inf_{f \in \mathcal{F}(W^{(0)}, R/\sqrt{m})} \frac{1}{|S|} \sum_{(x,y) \in S} \ell(f(x), y)$ with projected stochastic gradient descent. In this particular experiment, we set the width of MLPs for each layer as $\{200, 500, 2000\}$, respectively. Each model is trained for 200 epochs with a learning rate of $10^{-2}$. $R/\sqrt{m}$ varies from 0.1 to 1.9 with a step of 0.2. Figure 1 reports the average results of 10 runs.

2. Next, we investigate the impact of the clipping constant $C$ on the testing loss. To do so, we apply the DP-SGD optimizer to a three-layer MLP model with a width of 256, and train the model for 200 epochs with a learning rate of $\eta = 0.01$, using a training set of 60,000 samples. We set $\epsilon = 1$ and $\delta = 1/n^2$. The results are shown in Figure 2.

3. In Figure 3, we plot the mean testing error and 95% confidence interval based on 10 runs of the DP-SGD optimizer with different values of $m$. The optimizer is applied to a three-layer MLP model trained with a learning rate of 0.01 and 200 epochs, using $n = 5,000$, $\epsilon = 1$, $\delta = 1/n^2$, and $C = 20$.

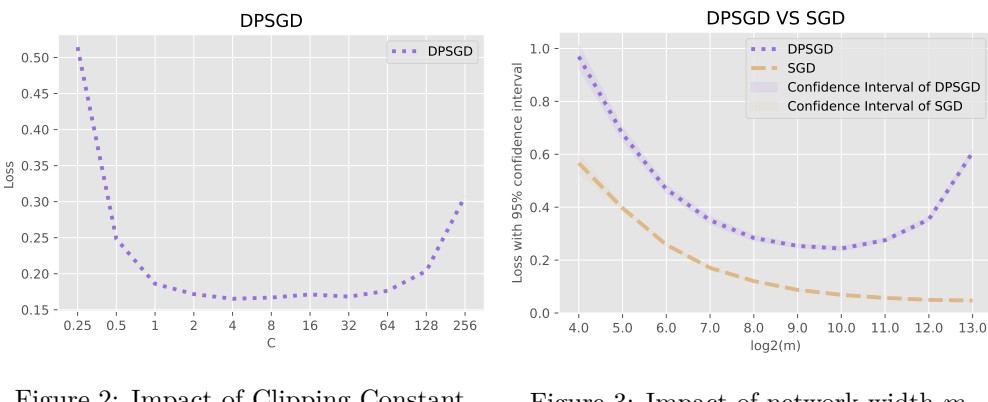

Figure 2: Impact of Clipping Constant.    Figure 3: Impact of network width $m$.

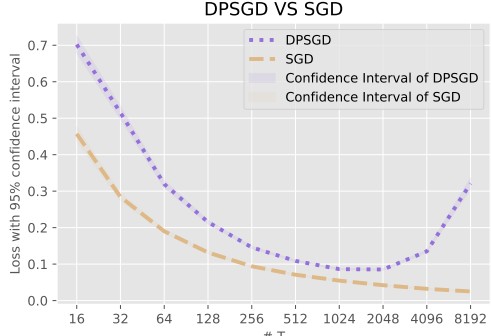 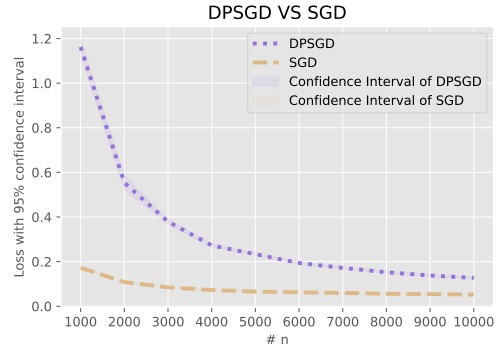

Figure 4: Impact of training iteration $T$.    Figure 5: Impact of sample size $n$.

4. In Figure 4, we plot the testing error mean value and 95% confidence interval of 10 runs with different value of $T$, which illustrates how the testing loss will change as $T$ increases with the setting of $n = 60,000$ and $m = 256$. Other parameters are the same as above.

5. In Figure 5, we plot the impact of n on the testing error of DP-SGD. In this setting, we choose the parameters satisfying the condition in Theorem 13, with $m = (n^{\frac{14}{15}})/2, T = 50n^{\frac{2}{15}}, \epsilon = 1, \delta = 1/n^2$ and $C = 20$. It is notable that our parameter setting is to make each term in the upper bound (7) decrease when $n$ becomes larger.

**Analysis.** The results presented in the above figures provide insightful findings on the impact of different hyperparameters on the excess population risk in DP-SGD-trained neural networks. Figure 1 shows that the approximation error in the upper bound of Theorem 13 yields a small and meaningful value, and that increasing the size of the hyperparameter space $R$ results in a smaller approximation error. Moreover, when the network width $m$ is increased, the approximation error tends to zero, indicating that the NTRF space can better fit wider neural networks on the training data. Figure 2 illustrates that when the clipping constant $C$ is chosen from $[1, 64]$, the excess population risk remains unaffected, which aligns with our theoretical analysis. The curves in Figure 3 and Figure 4 show that the excess population risk has a trade-off in choosing the network width $m$ and training iteration $T$, with neither of them being too large or too small. This is consistent with our theoretical findings and our discussions in Remark 15. Furthermore, Figure 5 demonstrates that the performance of DP-SGD is similar to that of non-private SGD when the sample size is sufficiently large. These findings highlight the importance of carefully tuning hyperparameters in DP-SGD-trained neural networks and provide valuable guidance for practical applications.

# 7 Conclusion

We presented a comprehensive study on the theoretical guarantees of DP Multi-layer Neural Networks. We started by considering the case where there are no hidden nodes, i.e., non-convex Generalized Linear Models. In the well-specified model, we studied the cases where the link function is Lipschitz and bounded (such as sigmoid) or unbounded (such as ReLU). We also analyzed ReLU regression in the misspecified model to highlight its difference from the well-specified model. Next, we extended our techniques to two-layer neural networks with sigmoid or ReLU activation functions in the well-specified model. Finally, we analyzed the standard DP-SGD method for general multi-layer neural networks and provided an upper bound for the excess population risk.

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

## A  Omitted Proofs in Section 4

**Proof of Lemma 2.** Note that by the definition of $\ell$ we have for any $w$,

$$\nabla\ell(w;x,y) = (\sigma(\langle w,x\rangle - y) \cdot x,$$
$$\nabla^2\ell(w;x,y) = \sigma'(\langle w,x\rangle \cdot xx^T,$$

where $\sigma'(\cdot)$ is a subgradient of $\sigma$. Since $\sigma$ is increasing, we have $\sigma'(\cdot) \geq 0$. Therefore we have $\nabla^2\ell(w;x,y) \succ 0$ and $\ell(\cdot;x,y)$ is convex and its Hessian matrix has rank at most 1. Since $\|\nabla\ell(w;x,y)\|_2 \leq 2BR$ and $\nabla^2\ell(w;x,y) \prec GR^2 I_d$, $\ell(\cdot;x,y)$ is $2B$-Lipschitz and $G$-smooth. □

**Proof of Lemma 3 .** For any fixed $x$ we have

$$\mathbb{E}_y[\ell(w;x,y)] - \mathbb{E}_y[\ell(w^*;x,y)] = \mathbb{E}_y \int_{\langle w^*,x\rangle}^{\langle w,x\rangle} (\sigma(z) - y)dz$$

$$= \int_{\langle w^*,x\rangle}^{\langle w,x\rangle} (\sigma(z) - \mathbb{E}_y y)dz = \int_{\langle w^*,x\rangle}^{\langle w,x\rangle} (\sigma(z) - \sigma(\langle w^*,x\rangle))dz$$

$$= \int_{\langle w^*,x\rangle}^{\langle w,x\rangle} \frac{\sigma'(z)(\sigma(z) - \sigma(\langle w^*,x\rangle))}{\sigma'(z)}dz$$

$$\geq \frac{1}{2G}(\sigma(\langle w,x\rangle) - \sigma(\langle w^*,x\rangle))^2,$$

where the last inequality is due to the fact that $\sigma$ is monotonically increasing and $G$-Lipschitz. Thus, taking the expectation of $x$ we have

$$L_\mathcal{P}^\ell(w) - L_\mathcal{P}^\ell(w^*) \geq \frac{1}{2G}\mathbb{E}_x(\sigma(\langle w,x\rangle) - \sigma(\langle w^*,x\rangle))^2$$

$$= \frac{1}{2G}(L_\mathcal{P}(w) - L_\mathcal{P}(w^*)).$$

$\square$

**Proof of Theorem 1.** Before the proof, we first provide some notations. For any $u, u' \in \mathbb{R}^d$, let $\|u\|_V = \sqrt{u^T VV^T u}$ as the semi-norm of $u$ induced by $V$, and let $\langle u, u'\rangle_V = u^T VV^T u'$.

Since $\ell(\cdot; x, y)$ is $2B$-Lipschitz and $G$-smooth, by Theorem 4.4 in Feldman et al. (2020) we can see it is $(\epsilon, \delta)$-DP when $\eta \leq \frac{2}{G}$. For utility, it is sufficient to show that

$$\mathbb{E}L_\mathcal{P}(w_k) - L_\mathcal{P}(w^*) \leq O(GBW(\frac{\sqrt{\theta \log \frac{1}{\delta}}}{n\epsilon} + \frac{1}{\sqrt{n}})). \tag{8}$$

Our proof follows the proof of convex GLM in Bassily et al. (2021). We first show the following lemma.

**Lemma 8.** *For each epoch $i$ we have $\mathbb{E}[L_\mathcal{P}^\ell(\bar{w}_i) - L_\mathcal{P}^\ell(\bar{w}_{i-1})] \leq \frac{\mathbb{E}\|\bar{w}_{i-1} - w_{i-1}\|_V^2}{2\eta_i n_i} + 2B^2\eta_i.$*

*Proof.* For simplicity we omit the subscript $i$ in $w_i^t$ and $\eta_i$. Denote $\Phi^t = \|w^t - \bar{w}_{i-1}\|_V^2$, we have

$$\Phi^{t+1} = \Phi^t - 2\eta\langle\nabla\ell(w^t; x^t, y^t), w^t - \bar{w}_{i-1}\rangle + \eta^2\|\nabla\ell(w^t; x^t, y^t)\|_2^2$$

$$\leq \Phi^t - 2\eta\langle\nabla\ell(w^t; x^t, y^t), w^t - \bar{w}_{i-1}\rangle + 4\eta^2 B^2,$$

where the first inequality is due to the fact that $\nabla\ell(w^t; x^t, y^t)$ in the span of $V$ and $\ell$ is $2B$-Lipschitz. Thus,

$$\langle\nabla\ell(w^t; x^t, y^t), w^t - \bar{w}_{i-1}\rangle \leq \frac{\Phi^t - \Phi^{t+1}}{2\eta} + 2B^2\eta.$$

By the convexity of $L_\mathcal{P}^\ell$ and take the expectation w.r.t all the data we have

$$\mathbb{E}[L_\mathcal{P}^\ell(w^t) - L_\mathcal{P}^\ell(\bar{w}_{i-1})] \leq \mathbb{E}[\langle\nabla L_\mathcal{P}^\ell(w^t), w^t - \bar{w}_{i-1}\rangle]$$

$$\leq \mathbb{E}[\frac{\Phi^t - \Phi^{t+1}}{2\eta}] + 2B^2\eta.$$

Thus, we have

$$\mathbb{E}[L_\mathcal{P}^\ell(\bar{w}_i) - L_\mathcal{P}^\ell(\bar{w}_{i-1})] \leq \frac{\mathbb{E}[\Phi^1]}{2\eta n_i} + 2B^2\eta = \frac{\mathbb{E}\|\bar{w}_{i-1} - w_{i-1}\|_V^2}{2\eta n_i} + 2B^2\eta.$$

$\square$

Now we back to our proof. Denote $\bar{w}_0 = w_\ell^*$ and $\zeta_0 = w_0 - w_\ell^*$, where $w_\ell^* = \arg\min_{w \in \mathbb{R}^d} L_\mathcal{P}^\ell(w)$, we have

$$\mathbb{E}[L_\mathcal{P}^\ell(w_k) - L_\mathcal{P}^\ell(w_\ell^*)]$$

$$= \mathbb{E}[L_\mathcal{P}^\ell(w_k) - L_\mathcal{P}^\ell(\bar{w}_k)] + \sum_{i=1}^k \mathbb{E}[L_\mathcal{P}^\ell(\bar{w}_i) - L_\mathcal{P}^\ell(\bar{w}_{i-1})]$$

$$\leq \sum_{i=1}^k \left(\frac{\mathbb{E}\|\zeta_{i-1}\|_V^2}{2\eta_i n_i} + 2B^2 \eta_i\right) + \mathbb{E}[L_\mathcal{P}^\ell(w_k) - L_\mathcal{P}^\ell(\bar{w}_k)].$$

Note that for all $2 \leq i \leq k$, we have

$$\mathbb{E}\|\zeta_{i-1}\|_V^2 = \mathbb{E}_V[\mathbb{E}_{\zeta_{i-1}}[\zeta_{i-1}^T V V^T \zeta_{i-1}|V]] \leq \theta \tau_{i-1}^2.$$

And when $i = 1$, $\mathbb{E}\|\zeta_{i-1}\|_V^2 \leq \|w_0 - w_\ell^*\|_2^2$. For $\mathbb{E}[L_\mathcal{P}^\ell(w_k) - L_\mathcal{P}^\ell(\bar{w}_k)]$ we have

$$\mathbb{E}[L_\mathcal{P}^\ell(w_k) - L_\mathcal{P}^\ell(\bar{w}_k)] \leq 2B\mathbb{E}[\langle \zeta_k, x \rangle] \leq 2B\tau_k \leq O\left(\frac{WB\sqrt{\log \frac{1}{\delta}}}{\epsilon n^{\frac{5}{2}}}\right).$$

In total we have

$$\sum_{i=1}^k \left(\frac{\mathbb{E}\|\zeta_{i-1}\|_V^2}{2\eta_i n_i} + 2B^2 \eta_i\right) + \mathbb{E}[L_\mathcal{P}^\ell(w_k) - L_\mathcal{P}^\ell(\bar{w}_k)]$$

$$\leq \sum_{i=2}^k \left(\frac{\theta \tau_{i-1}^2}{2\eta_i n_i} + 2B^2 \eta_i\right) + \frac{\|w_0 - w_\ell^*\|_2^2}{2\eta_1 n_1} + 2B^2 R^2 \eta_1 + O\left(\frac{WB\sqrt{\log \frac{1}{\delta}}}{\epsilon n^{\frac{5}{2}}}\right)$$

$$\leq O\left(BW\left(\frac{\sqrt{\theta \log \frac{1}{\delta}}}{n\epsilon} + \frac{1}{\sqrt{n}}\right)\right).$$

Note that by Lemma 3 we can see that $w_\ell^* = w^*$. Thus we have $\mathbb{E}L_\mathcal{P}(w) - L_\mathcal{P}(w^*) \leq 2G(\mathbb{E}L_\mathcal{P}^\ell(w) - L_\mathcal{P}^\ell(w^*)) \leq O(GBW(\frac{\sqrt{\theta \log \frac{1}{\delta}}}{n\epsilon} + \frac{1}{\sqrt{n}}))$. $\qquad \square$

**Proof of Theorem 2.** We first prove the privacy guarantee. Let $\alpha \leq 1$ be a parameter to be set later. From the JL property we know that with $m = O(\frac{\log n/\delta}{\alpha^2})$, then with probability at least $1 - \frac{\delta}{2}$ for all feature vectors we have $\|\Phi x_i\|_2 \leq (1 + \alpha)\|x_i\|_2 \leq 2\|x_i\|_2$, and $\|\Phi w^*\|_2 \leq 2\|w^*\|_2 \leq 2W$. In the following we will show if the previous events hold (which is denoted as $E$) then the Algorithm is $(\epsilon, \frac{\delta}{2})$-DP.

To see this note that $\ell(w_i^t; \tilde{x}_i^t, y_i^t) = \ell(w_i^t; \Phi x_i^t, y_i^t)$, we have $\|\nabla \ell(w_i^t; \tilde{x}_i^t, y_i^t)\|_2 = \|(\sigma(\langle w_i^t, \Phi x_i^t \rangle) - y_i^t)\Phi x_i^t\|_2 \leq 4B$ and $\ell(w_i^t; \tilde{x}_i^t, y_i^t)$ is $4G$-smooth. Thus, by Theorem 1 it is $(\epsilon, \frac{\delta}{2})$-DP if $\eta \leq \frac{2}{4G}$.

We then show the whole algorithm $\mathcal{A}$ is $(\epsilon, \delta)$-DP. Consider any event of the output $S$ and any neighboring datasets $D \sim D'$, we have

$$\mathcal{P}(\mathcal{A}(D) \in S) = \mathcal{P}(\mathcal{A}(D) \in S \bigcap E) + \mathcal{P}(\mathcal{A}(D) \in S \bigcap \bar{E})$$

$$\leq e^\epsilon \mathcal{P}(\mathcal{A}(D') \in S \bigcap E) + \frac{\delta}{2} + \mathcal{P}(\mathcal{A}(D) \in \bar{E})$$

$$\leq e^\epsilon \mathcal{P}(\mathcal{A}(D') \in S) + \delta.$$

Next we will show the utility. For simplicity we denote the projected distribution $\mathcal{P}' = \Phi\mathcal{P}$ we first decompose the excess population risk as the following:

$$\mathbb{E}[L_\mathcal{P}^\ell(\hat{w})] - L_\mathcal{P}^\ell(w^*) = \mathbb{E}[L_{\mathcal{P}'}^\ell(w_k)] - \mathbb{E}_{\Phi, \tilde{D}} L^\ell(\Phi w^*, \tilde{D}) + \mathbb{E}_{\Phi, \tilde{D}} L^\ell(\Phi w^*, \tilde{D}) - L_\mathcal{P}^\ell(w^*).$$

For the second term by Lemma 3 we know $w^*$ is also the global minimizer of $L_{\mathcal{P}}^\ell(w^*)$ and by Lemma 2 we know it is $G$-smooth, thus we have

$$\mathbb{E}_{\Phi,\tilde{D}} L^\ell(\Phi w^*, \tilde{D}) - L_{\mathcal{P}}^\ell(w^*) = \mathbb{E}_{\Phi,(x_i,y_i)\sim\mathcal{P}}[g^{y_i}(\langle\Phi w^*, \Phi x_i\rangle) - g^{y_i}(\langle w^*, x_i\rangle)]$$

$$\leq \nabla L_{\mathcal{P}}^\ell(w^*) + \frac{G}{2}\mathbb{E}\|\langle\Phi w^*, \Phi x_i\rangle - \langle w^*, x_i\rangle\|_2^2$$

$$= \frac{G}{2}\mathbb{E}\|\langle\Phi w^*, \Phi x_i\rangle - \langle w^*, x_i\rangle\|_2^2 \tag{9}$$

where the last inequality is due to that

$$\mathbb{E}_{x_i,\Phi}|\langle\Phi w^*, \Phi x_i\rangle - \langle w^*, x_i\rangle| = \mathbb{E}_{x_i}\mathbb{E}_\Phi|\langle\Phi w^*, \Phi x_i\rangle - \langle w^*, x_i\rangle|^2 \leq \tilde{O}(W\frac{1}{m}).$$

To bound the first term, we first $\Phi$ and use a similar analysis as in the proof of Theorem 1. The main difference here we use the following lemma instead of $\|\bar{w}_{i-1} - w_{i-1}\|_V^2$. Note that it is always true as $\|\bar{w}_{i-1} - w_{i-1}\|_V^2 \leq \|\bar{w}_{i-1} - w_{i-1}\|_2^2$

**Lemma 9.** *For each epoch $i$ we have* $\mathbb{E}[L_{\mathcal{P}'}^\ell(\bar{w}_i) - L_{\mathcal{P}'}^\ell(\bar{w}_{i-1})] \leq \frac{\mathbb{E}\|\bar{w}_{i-1}-w_{i-1}\|_2^2}{2\eta_i n_i} + 2B^2\eta_i.$

Denote $\bar{w}_0 = \Phi w^*$ and $\zeta_0 = w_0 - \Phi w^*$, we have

$$\mathbb{E}[L_{\mathcal{P}'}^\ell(w_k)] - \mathbb{E}L_{\mathcal{P}'}^\ell(\Phi w^*)$$

$$= \mathbb{E}[L_{\mathcal{P}'}^\ell(w_k) - L_{\mathcal{P}'}^\ell(\bar{w}_k)] + \sum_{i=1}^{k}\mathbb{E}[L_{\mathcal{P}'}^\ell(\bar{w}_i) - L_{\mathcal{P}'}^\ell(\bar{w}_{i-1})]$$

$$\leq \sum_{i=1}^{k}(\frac{\mathbb{E}\|\zeta_{i-1}\|_2^2}{2\eta_i n_i} + 2B^2\eta_i) + \mathbb{E}[L_{\mathcal{P}'}^\ell(w_k) - L_{\mathcal{P}'}^\ell(\bar{w}_k)].$$

Note that for all $2 \leq i \leq k$, we have $\mathbb{E}\|\zeta_{i-1}\|_2^2 = m\tau_{i-1}^2$. For $\mathbb{E}[L_{\mathcal{P}'}^\ell(w_k) - L_{\mathcal{P}'}^\ell(\bar{w}_k)]$ we have

$$\mathbb{E}[L_{\mathcal{P}'}^\ell(w_k) - L_{\mathcal{P}'}^\ell(\bar{w}_k)] \leq 2B\mathbb{E}[\langle\zeta_k, x\rangle] \leq 2B\tau_k \leq O(\frac{WB\sqrt{\log\frac{1}{\delta}}}{\epsilon n^{\frac{5}{2}}}).$$

In total we have

$$\sum_{i=1}^{k}(\frac{\mathbb{E}\|\zeta_{i-1}\|_2^2}{2\eta_i n_i} + 2B^2\eta_i) + \mathbb{E}[L_{\mathcal{P}'}^\ell(w_k) - L_{\mathcal{P}'}^\ell(\bar{w}_k)]$$

$$\leq \sum_{i=2}^{k}(\frac{m\tau_{i-1}^2}{2\eta_i n_i} + 2B^2\eta_i) + \frac{\|w_0 - \Phi w^*\|_2^2}{2\eta_1 n_1} + 2B^2R^2\eta_1 + O(\frac{WB\sqrt{\log\frac{1}{\delta}}}{\epsilon n^{\frac{5}{2}}})$$

$$\leq O(BW(\frac{\sqrt{m\log\frac{1}{\delta}}}{n\epsilon} + \frac{1}{\sqrt{n}})).$$

Note that the previous bound only holds when $\|\Phi x_i\| \leq (1 + \frac{\log\sqrt{n/\delta}}{m})$ which holds with probability at least $1 - \delta$. Thus we can use the same argument as in the Proof of Lemma 8 in Arora et al. (2022b) to transform the above result to a result of the expectation w.r.t $\Phi$ with an additional logarithmic factor. Thus, in total we have

$$\mathbb{E}[L_{\mathcal{P}}^\ell(\hat{w})] - L_{\mathcal{P}}^\ell(w^*) \leq \tilde{O}(GW\frac{1}{m} + BW(\frac{\sqrt{m\log\frac{1}{\delta}}}{n\epsilon} + \frac{1}{\sqrt{n}}))$$

Take $m = O(\log(n/\delta)(n\epsilon)^{\frac{2}{3}})$ we can get $\mathbb{E}L_{\mathcal{P}}(\hat{w}) - L_{\mathcal{P}}(w^*) \leq 2G(\mathbb{E}L_{\mathcal{P}}^\ell(\hat{w}) - L_{\mathcal{P}}^\ell(w^*)) \leq \tilde{O}(G^2WB(\frac{\sqrt{\log\frac{1}{\delta}}}{(n\epsilon)^{\frac{2}{3}}} + \frac{1}{\sqrt{n}})).$

$\square$

**Proof of Lemma 6.** Note that by our definition we have $\nabla f_\beta(w; x, y) = x g_\beta'^y(\langle w, x \rangle)$, where

$$g_\beta'^y(m) = \beta[m - \text{prox}_g^\beta(m)]. \tag{10}$$

Next we will provide an algorithm to approximate $\text{prox}_\ell^\beta(x)$ for given $x$. Recall that by the definition

$$\text{prox}_g^\beta(m) = \arg\min_{u \in \mathbb{R}} [g^y(u) + \frac{\beta}{2}|u - m|^2]$$

First we will show that $\text{prox}_g^\beta(m) \in [m - \frac{4B}{\beta}, m + \frac{4B}{\beta}]$. For simplicity we denote $u^* = \arg\min_{u \in \mathbb{R}} h(u) = \arg\min_{u \in \mathbb{R}} [g^y(u) + \frac{\beta}{2}|u - m|^2]$. Then since $u^*$ is the minimizer of $h(\cdot)$ we have

$$0 \leq h(m) - h(u^*) = g^y(m) - g^y(u^*) - \frac{\beta}{2}|u^* - m|^2$$

$$\iff \frac{\beta}{2}|u^* - m|^2 \leq g^y(m) - g^y(u^*).$$

Since we have $g^y(\cdot)$ is $2B$-Lipschitz (Lemma 2). Thus,

$$\frac{\beta}{2}|u^* - m|^2 \leq g^{(y)}(m) - g^{(y)}(u^*) \leq 2B|m - u^*| \iff |m - u^*| \leq \frac{4B}{\beta}.$$

That is

$$\text{prox}_g^\beta(m) = \arg\min_{u \in \mathbb{R}} [g^y(u) + \frac{\beta}{2}|u - m|^2] = \arg\min_{u \in \mathcal{Q}} [g^y(u) + \frac{\beta}{2}|u - m|^2],$$

where $\mathcal{Q} = [m - \frac{4B}{\beta}, m + \frac{4B}{\beta}]$. Moreover, on the constraint set $\mathcal{Q}$, function $h(\cdot)$ is $\beta$-strongly convex and $2B + 4B = 6B$-Lipschitz. Thus, from a standard result on convergence of Gradient Decent for strongly and Lipschitz functions (which corresponds to Step 3 to 7 in Algorithm 4, note that Step 5 is just the projection onto the set $\mathcal{Q}$) in Bubeck et al. (2015) we can see that after $T$-steps we have

$$\frac{\beta}{2}|\hat{w} - u^*|^2 \leq h(\hat{w}) - h(u^*) \leq \frac{72B^2}{\beta(T+1)}.$$

Thus we have $|\hat{w} - u^*| \leq \frac{12B}{\beta\sqrt{T}}$. Thus we have $\|x\beta[\langle w, x \rangle - \hat{w}] - \nabla f_\beta(w; x, y)\|_2 \leq \frac{12B}{\sqrt{T}}$.

$\square$

**Proof of Theorem 4.** We first proof the guarantee of $(\epsilon, \delta)$-DP. Note that unlike Algorithm 2, here we use an approximation of $\nabla f_\beta(w; x, y)$. Consider a neighboring dataset $D'$ of $D$ assume the different samples are in $D_i$ which are denoted as $x_i^t$ and $x_i'^t$ respectively. Moreover, we denote $w_i'^t$ as the parameters when implementing the algorithm on $D'$. Then by our assumption we have $w_i^t = w_i'^t$ but $w_i^{t_0} \neq w_i'^{t_0}$ when $t_0 \geq t+1$. Moreover, for any $t_0 \geq t+1$ we have

$$\|w_i^{t_0} - w_i'^{t_0}\|_2 \leq \|w_i^{t_0-1} - \eta_i \nabla f_\beta(w_i^{t_0-1}; x_i^{t_0-1}, y_i^{t_0-1}) - w_i'^{t_0-1} + \eta_i \nabla f_\beta(w_i^{t_0-1}; x_i'^{t_0-1}, y_i'^{t_0-1})\|_2 + 2\eta_i\gamma$$

where $x_i^{t_0} \neq x_i'^{t_0}$ if $t_0 = t+1$ and $x_i^t = x_i'^t$ otherwise. Note that since $f_\beta$ is $\beta$-smooth. Thus, by using a similar proof as in Hardt et al. (2016) we have when $\eta \leq \frac{2}{\beta}$ and $t_0 \geq t+2$ we always have

$$\|w_i^{t_0-1} - \eta_i \nabla f_\beta(w_i^{t_0-1}; x_i^{t_0-1}, y_i^{t_0-1}) - w_i'^{t_0-1} + \eta_i \nabla f_\beta(w_i^{t_0-1}; x_i'^{t_0-1}, y_i'^{t_0-1})\|_2 \leq \|w_i^{t_0-1} - w_i'^{t_0-1}\|_2.$$

Thus, we always have $\|w_i^{t_0} - w_i'^{t_0}\|_2 \leq \|w_i^{t_0-1} - w_i'^{t_0-1}\|_2 + 2\eta_i\gamma$.

When $t_0 = t+1$ by using a similar proof as in Hardt et al. (2016) ans since $f_\beta$ is $2B$-Lipschitz we have $\|w_i^{t_0-1} - \eta_i \nabla f_\beta(w_i^{t_0-1}; x_i^{t_0-1}, y_i^{t_0-1}) - w_i'^{t_0-1} + \eta_i \nabla f_\beta(w_i^{t_0-1}; x_i'^{t_0-1}, y_i'^{t_0-1})\|_2 \leq 4B\eta_i$. Thus we have $\|w_i^{t+1} - w_i'^{t+1}\|_2 \leq 4B\eta_i + 2\gamma\eta_i$.

In total we have $\|w_i^{t_0} - w_i'^{t_0}\|_2 \leq 4B\eta_i + 2\eta_i\gamma t_0$. And thus $\|\bar{w}_i - \bar{w}_i'\| \leq 4B\eta_i + \gamma\eta_i(n+1) \leq 5B\eta_i$. Thus, the $\ell_2$-norm sensitivity is $5B\eta_i$. By the Gaussian mechanism we have the algorithm is $(\epsilon, \delta)$-DP.

Next we will focus on the utility. We first show the following lemma which follows Bassily et al. (2021) for self-completeness:

**Lemma 10.** *Let $\alpha, \eta$ be as in Theorem 4. Then for each phase $i$, we have*

$$\mathbb{E}[F_\beta(\bar{w}_i) - F_\beta(\bar{w}_{i-1})] \leq \frac{\mathbb{E}[\|w_{i-1} - \bar{w}_{i-1}\|_V^2]}{2\eta_i n_i} + \frac{5\eta_i B^2}{2} + \frac{(B\mathbb{E}[\|w_{i-1} - \bar{w}_{i-1}\|_V] + 1)}{\sqrt{n}\log n}.$$

*Proof.* For simplicity we denote $F_\beta(w) = \mathbb{E}[f_\beta(w; x, y)]$ omit the subscript $i$. Denote $\Phi^t = \|w^t - \bar{w}_{i-1}\|_V^2$, we have

$$\Phi^{t+1} = \Phi^t - 2\eta\langle\tilde{\nabla}f_\beta(w^t; x^t, y^t), w^t - \bar{w}_{i-1}\rangle_V + \eta^2\|\tilde{\nabla}f_\beta(w^t; x^t, y^t)\|_V^2$$
$$\leq \Phi^t - 2\eta\langle\nabla f_\beta(w^t; x^t, y^t), w^t - \bar{w}_{i-1}\rangle + 2\eta\gamma\|w^t - \bar{w}_{i-1}\|_V + \eta^2(\gamma^2 + 4B^2),$$

where the first inequality is due to the fact that $\nabla f_\beta(w^t; x^t, y^t)$ in the span of $V$. Thus,

$$\langle\nabla f_\beta(w^t; x^t, y^t), w^t - \bar{w}_{i-1}\rangle \leq \frac{\Phi^t - \Phi^{t+1}}{2\eta} + \gamma\|w^t - \bar{w}_{i-1}\|_V + \frac{\eta}{2}(\gamma^2 + 4B^2).$$

Taking the expectation w.r.t all randomness we have

$$\langle\nabla F_\beta(w^t), w^t - \bar{w}_{i-1}\rangle \leq \frac{\mathbb{E}[\Phi^t - \Phi^{t+1}]}{2\eta} + \gamma\mathbb{E}\|w^t - \bar{w}_{i-1}\|_V + \frac{\eta}{2}(\gamma^2 + 4B^2).$$

By the convexity of $F_\beta$ we have $\langle\nabla F_\beta(w^t), w^t - \bar{w}_{i-1}\rangle \geq \mathbb{E}[F_\beta(w^t) - F_\beta(\bar{w}_{i-1})]$. Thus, we have

$$\mathbb{E}[F_\beta(\bar{w}_i) - F_\beta(\bar{w}_{i-1})] \leq \frac{\mathbb{E}[\Phi^1]}{2\eta n_i} + \frac{\gamma}{n_i}\mathbb{E}[\sum_{t=1}^{n_i}\|w^t - \bar{w}_{i-1}\|_V] + \frac{\eta}{2}(\gamma^2 + 4B^2).$$

Next we bound the term $\sum_{t=1}^{n_i}\|w^t - \bar{w}_{i-1}\|_V$:

$$\|w^t - \bar{w}_{i-1}\|_V \leq \|w^{t-1} - \bar{w}_{i-1}\|_V + \|w^t - w^{t-1}\|_V$$
$$\leq \cdots \leq \|w_{i-1} - \bar{w}_{i-1}\|_V + \sum_{j=2}^{t}\|w^j - w^{j-1}\|_V$$
$$\leq \sqrt{\Phi^1} + \eta(t-1)(2B + \gamma).$$

In total we have

$$\mathbb{E}[F_\beta(\bar{w}_i) - F_\beta(\bar{w}_{i-1})]$$
$$\leq \frac{\mathbb{E}[\Phi^1]}{2\eta n_i} + \alpha\mathbb{E}[\sqrt{\Phi^1} + \eta n_i(2B + \gamma)] + \frac{\eta}{2}(\gamma^2 + 4B^2)$$
$$\leq \frac{\mathbb{E}[\Phi^1]}{2\eta n_i} + \frac{5\eta B^2}{2} + \gamma(\mathbb{E}[\sqrt{\Phi^1}] + 3n_i\eta B),$$

where the last step follows from the fact that $\gamma = \frac{B}{n\log n} \leq B$. Since we have $\eta \leq \frac{W}{6B\sqrt{n}}$, we have $3n_i\eta B \leq \sqrt{n}$. In total we have

$$\mathbb{E}[F_\beta(\bar{w}_i) - F_\beta(\bar{w}_{i-1})] \leq \frac{\mathbb{E}[\Phi^1]}{2\eta n_i} + \frac{5\eta B^2}{2} + \frac{(B\mathbb{E}[\sqrt{\Phi^1}] + 1)}{\sqrt{n}\log n}.$$

$\square$

Now we back to the proof of Theorem 4. Denote $\bar{w}_0 = w_\beta^*$ and $\zeta_0 = w_0 - w^*$ and by Lemma 10 we have

$$\mathbb{E}[F_\beta(w_k) - F_\beta(w^*)]$$

$$= \sum_{i=1}^{k} \mathbb{E}[F_\beta(\bar{w}_k) - F_\beta(\bar{w}_{k-1})] + \mathbb{E}[F_\beta(w_k) - F_\beta(\bar{w}_k)]$$

$$\leq \sum_{i=1}^{k} \frac{\mathbb{E}[\|w_{i-1} - \bar{w}_{i-1}\|_V^2]}{2\eta_i n_i} + \frac{5\eta_i B^2}{2} + \frac{(B\mathbb{E}[\|w_{i-1} - \bar{w}_{i-1}\|_V] + 1)}{\sqrt{n} \log n}$$

$$+ \mathbb{E}[F_\beta(w_k) - F_\beta(\bar{w}_k)]$$

$$= \sum_{i=1}^{k} \frac{\mathbb{E}[\|\zeta_{i-1}\|_V^2]}{2\eta_i n_i} + \frac{5\eta_i B^2}{2} + \frac{B\mathbb{E}[\|\zeta_{i-1}\|_V] + 1}{\sqrt{n} \log n} + \mathbb{E}[F_\beta(w_k) - F_\beta(\bar{w}_k)].$$

Note that for all $2 \leq i \leq k$, we have

$$\mathbb{E}\|\zeta_{i-1}\|_V^2 = \mathbb{E}_V[\mathbb{E}_{\zeta_{i-1}}[\zeta_{i-1}^T V V^T \zeta_{i-1} | V]] \leq \theta \tau_{i-1}^2.$$

And when $i = 1$, $\mathbb{E}\|\zeta_{i-1}\|_V^2 \leq \|w_0 - w_\ell^*\|_2^2$. For $\mathbb{E}[F_\beta(w_k) - F_\beta(\bar{w}_k)]$ we have

$$\mathbb{E}[F_\beta(w_k) - F_\beta(\bar{w}_k)] \leq 2BR\mathbb{E}[\langle \zeta_k, x \rangle] \leq 2B\tau_k \leq O(\frac{WBR\sqrt{\log \frac{1}{\delta}}}{\epsilon n^{\frac{5}{2}}}).$$

In total we have

$$\mathbb{E}[F_\beta(w_k) - F_\beta(w^*)] \leq \sum_{i=1}^{k} \frac{\mathbb{E}[\|\zeta_{i-1}\|_V^2]}{2\eta_i n_i} + \frac{5\eta_i B^2}{2} + \frac{B\mathbb{E}[\|\zeta_{i-1}\|_V] + 1}{\sqrt{n} \log n} + \mathbb{E}[F_\beta(w_k) - F_\beta(\bar{w}_k)]$$

$$\leq O(B(\|w_0 - w^*\|_2^2 + 1)(\frac{\sqrt{\theta \log \frac{1}{\delta}}}{n\epsilon} + \frac{1}{\sqrt{n}}))$$

$$+ \sum_{i=2}^{k} [\frac{\theta \tau_{i-1}^2}{2\eta_i n_i} + \frac{5\eta_i B^2}{2} + \frac{B(\sqrt{\theta}\tau_{i-1} + 1)}{\sqrt{n} \log n}] + O(\frac{WB\sqrt{\log \frac{1}{\delta}}}{\epsilon n^{\frac{5}{2}}})$$

$$\leq O(W^2 B(\frac{\sqrt{\theta \log \frac{1}{\delta}}}{n\epsilon} + \frac{1}{\sqrt{n}})).$$

Thus, by Lemma 4 we have

$$\mathbb{E}[L_{\mathcal{P}}](w_k) - L_{\mathcal{P}}(w^*) \leq 2G(\mathbb{E}L_{\mathcal{P}}^\ell(w) - L_{\mathcal{P}}^\ell(w^*))$$

$$\leq O(GW^2 B(\frac{\sqrt{\theta \log \frac{1}{\delta}}}{n\epsilon} + \frac{1}{\sqrt{n}}) + \frac{GB^2}{\beta}).$$

Take $\beta = O(\frac{\sqrt{n}B}{W^2})$ we can get the result. $\qquad \square$

**Proof of Theorem 5.** The proof has the same idea of the proof in Theorem 2. And here we use the proof of Theorem 4 instead of Theorem 1. For simplicity we omit it here. $\qquad \square$

**Proof of Theorem 6.** First we will show the $(\epsilon, \delta)$-DP guarantee. Similar to the proof of Theorem 2 we know that when $k = O(\frac{\log n/\delta}{\alpha^2})$ with some $\alpha \leq 1$ we have with probability at least $1 - \frac{\delta}{2}$, $\|\Phi x_i\|_2 \leq (1 + \alpha)\|x_i\|_2 \leq 2$ and $\|\Phi w^*\|_2 \leq 2W$. Under this event we can easily calculate the $\ell_2$-norm sensitivity of $\frac{1}{n} \sum_{i=1}^{n} (\max\{0, \langle \tilde{w}_t, \Phi x_i \rangle\} - y_i)\Phi x_i$, which is $\frac{4(4W+B)}{n}$. Thus the line 2-4 is $(\epsilon, \frac{\delta}{2})$-DP and the whole algorithm is $(\epsilon, \delta)$-DP.

Next we will show the utility, note that line 2-4 is equivalent to using the projected gradient descent to $L^\ell(w; \tilde{D})$ with $\ell(w; x, y) = \int_0^{\langle w,x \rangle} (\sigma(z) - y)dz$. Then denote $\mathcal{P}' = \Phi\mathcal{P}$ and $\tilde{w} = \frac{\sum_{t=1}^T \tilde{w}_t}{T}$ we have

$$\mathbb{E}L_\mathcal{P}^\ell(\bar{w}) - L_\mathcal{P}^\ell(w^*) = [\mathbb{E}L_{\mathcal{P}'}^\ell(\tilde{w})) - \mathbb{E}L^\ell(\Phi w^*, \tilde{D})] + [\mathbb{E}L^\ell(\Phi w^*, \tilde{D})] - L_\mathcal{P}^\ell(w^*)]$$
$$\leq [L_{\mathcal{P}'}^\ell(\tilde{w})) - \min_{w \in \tilde{W}} L_{\mathcal{P}'}^\ell(w)] + [L_{\mathcal{P}'}^\ell(\tilde{w}_{T+1})) - L_\mathcal{P}^\ell(w^*)].$$

For the second term due to Lemma 3 we known $w^*$ is the global minimizer of $L_\mathcal{P}^\ell(w)$ and thus $\nabla L_\mathcal{P}^\ell(w^*) = 0$, moreover we can see $\ell$ is 1-smooth, thus we have

$$\mathbb{E}L^\ell(\Phi w^*, \tilde{D}) - L_\mathcal{P}^\ell(w^*) \leq \frac{1}{2}\mathbb{E}[|\langle \Phi w^*, \Phi x \rangle - \langle w^*, x \rangle|^2]$$
$$\leq O(W\alpha^2) = O(\frac{W \log n/\delta}{m}).$$

For the first term, we have

$$\mathbb{E}L_{\mathcal{P}'}^\ell(\tilde{w})) - \mathbb{E}L^\ell(\Phi w^*, \tilde{D}) = \mathbb{E}L_{\mathcal{P}'}^\ell(\tilde{w}) - \mathbb{E}L_{\mathcal{P}'}^\ell(\Phi w^*) \leq \mathbb{E}L_{\mathcal{P}'}^\ell(\tilde{w})) - \min_{w \in \tilde{W}} L_{\mathcal{P}'}^\ell(w)$$
$$= \mathbb{E}L_{\mathcal{P}'}^\ell(\tilde{w})) - \mathbb{E}L^\ell((\tilde{w}, \tilde{D}) + \mathbb{E}L^\ell((\tilde{w}, \tilde{D}) - \mathbb{E}L^\ell((\tilde{w}^*, \tilde{D})$$

Since $\ell(w; \Phi x_i, y_i)$ is a $2(4W + B)$-Lipschitz and 4-smooth function and the algorithm is just the PGD for the empirical risk function. The first term, which is the generalization error is bounded by Lipschitz times the stability i.e., $O((W + B)^2 \frac{\eta T}{n})$. The second term is bounded by the excess empirical risk, we have

$$\mathbb{E}[\|\tilde{w}_{t+1} - \tilde{w}^*\|_2^2] \leq \mathbb{E}\|\tilde{w}_t - \tilde{w}^* - \eta(\nabla L^\ell(\tilde{w}_t, \tilde{D}) + \eta_t)\|_2^2$$
$$\leq \mathbb{E}\|\tilde{w}_t - \tilde{w}^*\|_2^2 + \eta^2\|\nabla L^\ell(\tilde{w}_t, \tilde{D})\|_2^2 + \eta^2\sigma^2 m - 2\eta(L^\ell(\tilde{w}_t, \tilde{D}) - L^\ell(\tilde{w}^*, \tilde{D}))$$
$$\leq \mathbb{E}\|\tilde{w}_t - \tilde{w}^*\|_2^2 + 4\eta^2(4W + B)^2 + \eta^2\sigma^2 m - 2\eta(L^\ell(\tilde{w}_t, \tilde{D}) - L^\ell(\tilde{w}^*, \tilde{D}))$$

Thus, taking the sum for $t = 1, \cdots, T$ we have

$$L^\ell(\tilde{w}, \tilde{D}) - L^\ell(\tilde{w}^*, \tilde{D}) \leq \frac{\mathbb{E}\|\tilde{w}_1 - \tilde{w}^*\|_2^2}{2\eta T} + 4\eta(4W + B)^2 + O(\frac{\eta(W + B)^2 m T \log(1/\delta)}{n^2 \epsilon^2}).$$

In total we have

$$\mathbb{E}L_{\mathcal{P}'}^\ell(\tilde{w})) - \mathbb{E}L^\ell(\Phi w^*, \tilde{D}) \leq O(\frac{W^2}{\eta T} + \frac{\eta(W + B)^2 T m \log\frac{1}{\delta}}{n^2 \epsilon^2} + (W + B)^2\frac{\eta G T}{n} + \eta(W + B)^2).$$

Note that the previous bound only holds when $\|\Phi x_i\| \leq (1 + \frac{\log\sqrt{n/\delta}}{m})$ which holds with probability at least $1 - \delta$. Thus we can use the same argument as in the Proof of Lemma 8 in Arora et al. (2022b) to transform the above result to a result of the expectation w.r.t $\Phi$ with an additional logarithmic factor. Thus we have

$$\mathbb{E}L_\mathcal{P}^\ell(w_{T+1}) - L_\mathcal{P}^\ell(w^*) \leq \tilde{O}(\frac{W^2}{\eta T} + \frac{\eta(W + B)^2 T m \log\frac{1}{\delta}}{n^2 \epsilon^2} + (W + B)^2\frac{\eta T}{n} + \frac{W \log n/\delta}{m} + \eta(W + B)^2).$$

Thus, when take $\eta = \frac{W}{\sqrt{T}\max\{(W+B), \frac{(W+B)\sqrt{mT \log(1/\delta)}}{n\epsilon}\}} \leq \frac{W}{(W+B)\sqrt{T}} \leq \frac{1}{2}$ and $T = O(\min\{n, \frac{n^2\epsilon^2}{m \log 1/\delta}\})$ we have

$$\mathbb{E}L_\mathcal{P}^\ell(w_{T+1}) - L_\mathcal{P}^\ell(w^*) \leq O(\frac{W(W + B)}{\sqrt{n}} + \frac{W(W + B)\sqrt{m \log 1/\delta}}{n\epsilon} + \frac{W \log n/\delta}{m}).$$

Take $m = O((n\epsilon)^{\frac{2}{3}} \log n/\delta)$ we can get the result. □

**Proof of Theorem 7.** We first show the proof of privacy. Note that since each iteration we use one data $D_i$. Thus, it is sufficient to show the algorithm is $(\epsilon, \delta)$-DP in the $i$-th iteration with fixed $w_{i-1}$. This is true since the $\ell_2$-norm sensitivity of $\nabla L^\ell(w_{i-1}; D_i)$ is $\frac{\sqrt{d}\|w_{i-1}\|_2 + B}{m}$ based on our assumption.

Next we will focus on the utility. By the concentration property of Gaussian distribution we know that with probability at least $1 - \zeta$, we have $\|\zeta_{i-1}\|_2^2 \leq O(d\frac{(\sqrt{d}\|w_{i-1}\|_2 + B)^2 \log \frac{1}{\zeta} \log \frac{1}{\delta}}{m^2 \epsilon^2})$. Thus, with probability at least $1 - \zeta$, $\|\zeta_{i-1}\|_2^2 \leq O(d\frac{(\sqrt{d}\|w_{i-1}\|_2 + B)^2 \log \frac{T}{\zeta} \log \frac{1}{\delta}}{m^2 \epsilon^2})$ for all $i = 1, \cdots, T$. Below we will always assume this event holds.

Before our proof we first recall the following lemmas:

**Lemma 11** (Corollary 2.4 of Diakonikolas et al. (2020)). *If $\mathcal{P}_\mathcal{X}$ is isotropic, then for any vector $w$, the distance between $\chi_\mathcal{P}^{\sigma_w}$ and $\chi_\mathcal{P}$ is bounded by $\sqrt{L_\mathcal{P}(w)}$, i.e., $\|\chi_\mathcal{P}^{\sigma_w} - \chi_\mathcal{P}\|_2 \leq \sqrt{L_\mathcal{P}(w)}$.*

**Lemma 12** (Lemma 4.2 in Diakonikolas et al. (2020)). *Under Assumption 2 , we have $\mathbb{E}_\mathcal{P}[(\sigma(\langle w, x \rangle) - \sigma(\langle w^*, x \rangle))^2] \leq \mu(\|\chi_\mathcal{P}^{\sigma_w} - \chi_\mathcal{P}^{\sigma_{w^*}}\|_2^2)$ with some constant $\mu$.*

Thus in total we have

$$
\begin{aligned}
\mathbb{E}_\mathcal{P}[(\sigma(\langle w, x \rangle) - \sigma(\langle w^*, x \rangle))^2] &\leq \mu(\|\chi_\mathcal{P}^{\sigma_w} - \chi_\mathcal{P}^{\sigma_{w^*}}\|_2^2) \\
&\leq 2\mu(\|\chi_\mathcal{P}^{\sigma_w} - \chi_\mathcal{P}\|_2^2 + \|\chi_\mathcal{P}^{\sigma_{w^*}} - \chi_\mathcal{P}\|_2^2) \\
&\leq 2\mu L_\mathcal{P}(w^*) + 2\mu\|\nabla L_\mathcal{P}^\ell(w)\|_2.
\end{aligned}
\tag{11}
$$

On the other side by the triangle inequality we have

$$
L_\mathcal{P}(w) \leq 2L_\mathcal{P}(w^*) + 2\mathbb{E}_\mathcal{P}[(\sigma(\langle w, x \rangle) - \sigma(\langle w^*, x \rangle))^2].
$$

In total we have

$$
L_\mathcal{P}(w) \leq 2(1 + 2\mu)L_\mathcal{P}(w^*) + 4\mu\|\nabla L_\mathcal{P}^\ell(w)\|_2.
$$

In the following we will bound the term of $\|\nabla L_\mathcal{P}^\ell(w_T)\|_2$ in Algorithm 8. We recall the following lemmas in Diakonikolas et al. (2020).

**Lemma 13.** *Consider a ball $B(0, r)$ with radius $r$, under Assumption 2, if $\sigma$ is the sigmod link function. Then as long as*

$$
n \geq \tilde{\Omega}(\frac{d}{\alpha^2} \log^4 \frac{d}{\zeta}(r + 1)^2),
$$

*we have for fixed $w \in B(0, r)$*

$$
\|\frac{1}{n}\sum_{i=1}^n (\sigma(\langle w, x_i \rangle) - y_i)x_i\|_2 \leq \alpha.
$$

**Lemma 14.** *As long as $\alpha \leq \|w_\ell^*\|_2$, $\zeta \geq \exp(-O(\sqrt{d}))$ and*

$$
n \geq \tilde{\Omega}(\frac{d}{\mu^2} \log \frac{\|w_\ell^*\|_2 + 1}{\mu\zeta}),
$$

*with probability at least $1 - \zeta$ we have for all $w$ such that $\frac{\alpha}{3} \leq \|w - w_\ell^*\|_2 \leq 2\|w_\ell^*\|_2$,*

$$
\langle \nabla L^\ell(w; D) - \nabla L^\ell(w_\ell^*; D), w - w_\ell^* \rangle \geq \tau\|w - w_\ell^*\|_2^2 + \beta\|\nabla L^\ell(w; D) - \nabla L^\ell(w_\ell^*; D)\|_2^2
$$

*with $\tau = \frac{\mu}{3}$ and $\beta = \frac{1}{8}$.*

Now lets back to our proof, we will first show that $\|w_i - w_\ell^*\|_2 \leq 2\|w_\ell^*\|_2$ for all $i = 0, \cdots, T$ when $n$ is large enough:

**Lemma 15.** *Suppose the event of $\|\zeta_{i-1}\|_2^2 \leq O(\frac{(\sqrt{d}\|w_{i-1}\|_2 + B)^2 \log \frac{T}{\zeta} \log \frac{1}{\delta}}{m^2 \epsilon^2})$ for all $i = 1, \cdots, T$ holds, then we have $\|w_i - w_\ell^*\|_2 \leq 2\|w_\ell^*\|_2$ for all $i = 0, \cdots, T$ when $m \geq \tilde{\Omega}(\sqrt{\frac{1}{\tau}(4\eta + \frac{1}{\tau})}\frac{(\sqrt{d}\|w_\ell^*\|_2 + B)\sqrt{\log 1/\delta \log 1/\zeta}}{\epsilon\alpha})$*

*Proof.* We will show it by using induction. This is true for $i = 0$ since $w_0 = 0$. Suppose this is true for some $i - 1$, then our goal is to show $\|w_i - w^*\|_2 \leq 2\|w_\ell^*\|_2$. We consider two cases.

The first case is $2\|w_\ell^*\|_2 \geq \|w_{i-1} - w_\ell^*\| \geq \frac{\alpha}{3}$. Then we have

$$
\begin{aligned}
\|w_i - w_\ell^*\|_2^2 &= \|w_{i-1} - w_\ell^*\|_2^2 - \eta\langle \nabla L^\ell(w_{i-1}; D_i) + \zeta_{i-1}, w_{i-1} - w_\ell^*\rangle + \eta^2\|\nabla L^\ell(w_{i-1}; D_i) + \zeta_{i-1}\|_2^2 \\
&= \|w_{i-1} - w_\ell^*\|_2^2 - \eta\langle \nabla L^\ell(w_{i-1}; D_i) - \nabla L^\ell(w_\ell^*; D_i), w_{i-1} - w_\ell^*\rangle \\
&\quad + \eta^2\|\nabla L^\ell(w_{i-1}; D_i) + \zeta_{i-1}\|_2^2 - \eta\langle \nabla L^\ell(w_\ell^*; D_i) + \zeta_{i-1}, w_{i-1} - w_\ell^*\rangle \\
&\leq (1 - \tau\eta)\|w_{i-1} - w_\ell^*\|_2^2 - \eta\beta\|\nabla L^\ell(w_{i-1}; D_i) - \nabla L^\ell(w_\ell^*; D_i)\|_2^2 + 2\eta^2\|\nabla L^\ell(w_{i-1}; D_i) - \nabla L^\ell(w_\ell^*; D_i)\|_2^2 \\
&\quad + 2\eta^2\|\nabla L^\ell(w_\ell^*; D_i) + \zeta_{i-1}\|_2^2 - \eta\langle \nabla L^\ell(w_\ell^*; D_i) + \zeta_{i-1}, w_{i-1} - w_\ell^*\rangle \\
&\leq (1 - \tau\eta)\|w_{i-1} - w_\ell^*\|_2^2 - \eta(\beta - 2\eta)\|\nabla L^\ell(w_{i-1}; D_i) - \nabla L^\ell(w_\ell^*; D_i)\|_2^2 + 4\eta^2\|\nabla L^\ell(w_\ell^*; D_i)\|_2^2 + 4\eta^2\|\zeta_{i-1}\|_2^2 \\
&\quad + \frac{\tau\eta}{2}\|w_{i-1} - w_\ell^*\|_2^2 + \frac{\eta}{\tau}\|\nabla L^\ell(w_\ell^*; D_i)\|_2^2 + \frac{\eta}{\tau}\|\zeta_{i-1}\|_2^2 \\
&\leq (1 - \frac{\tau\eta}{2})\|w_{i-1} - w_\ell^*\|_2^2 + \eta(4\eta + \frac{1}{\tau})\|\nabla L^\ell(w_\ell^*; D_i)\|_2^2 + \eta(4\eta + \frac{1}{\tau})\|\zeta_{i-1}\|_2^2 \quad (12) \\
&= (1 - \frac{\tau\eta}{2})\|w_{i-1} - w_\ell^*\|_2^2 + \eta(4\eta + \frac{1}{\tau})\|\nabla L^\ell(w_\ell^*; D_i)\|_2^2 + \tilde{O}(d\eta(4\eta + \frac{1}{\tau})\frac{(\sqrt{d}\|w_{i-1}\|_2 + B)^2 \log(1/\delta)\log 1/\zeta}{m^2\epsilon^2}) \\
&\leq (1 - \frac{\tau\eta}{2} + \tilde{O}(\eta(4\eta + \frac{1}{\tau})\frac{d^2\log(1/\delta)\log 1/\zeta}{m^2\epsilon^2}))\|w_{i-1} - w_\ell^*\|_2^2 + \eta(4\eta + \frac{1}{\tau})\|\nabla L^\ell(w_\ell^*; D_i)\|_2^2 \\
&\quad + \tilde{O}(d\eta(4\eta + \frac{1}{\tau})\frac{(\sqrt{d}\|w_\ell^*\|_2 + B)^2 \log(1/\delta)\log 1/\zeta}{m^2\epsilon^2}), \quad (13)
\end{aligned}
$$

where the first inequality is due to Lemma 14. Thus, we can see that when

$$
m \geq \tilde{\Omega}(\sqrt{\frac{\eta}{\tau} + \frac{1}{\tau^2}}\frac{d\sqrt{\log 1/\delta \log 1/\zeta}}{\epsilon}),
$$

take $\alpha = \sqrt{\frac{\tau}{9(4\tau\beta+1)}\alpha^2}$ in Lemma 13 and when $m \geq \tilde{\Omega}(\sqrt{d\frac{1}{\tau}(4\eta + \frac{1}{\tau})}\frac{(\sqrt{d}\|w_\ell^*\|_2 + B)\sqrt{\log 1/\delta \log 1/\zeta}}{\epsilon\alpha})$. Then we have

$$
\|w_i - w_\ell^*\|_2^2 \leq (1 - \frac{\tau\eta}{4})\|w_{i-1} - w_\ell^*\|_2^2 + \frac{2\alpha^2}{9} \leq 4\|w_\ell^*\|_2^2.
$$

We then consider case 2 where $\|w_{i-1} - w_\ell^*\|_2 \leq \frac{\alpha}{3}$. Then we have

$$
\begin{aligned}
\|w_i - w_\ell^*\|_2 &= \|w_{i-1} - w_\ell^* - \eta(\nabla L^\ell(w_{i-1}; D_i) + \zeta_{i-1}\|_2 \\
&\leq \|w_{i-1} - w_\ell^* - \eta(\nabla L^\ell(w_{i-1}; D_i) - \nabla L^\ell(w_\ell^*; D_i)\|_2 + \eta\|\zeta_{i-1}\|_2 + \eta\|\nabla L^\ell(w_\ell^*; D_i)\|_2 \\
&\leq \|w_{i-1} - w_\ell^*\|_2 + \eta\|\zeta_{i-1}\|_2 + \eta\|\nabla L^\ell(w_\ell^*; D_i)\|_2 \leq \alpha \leq \|w_\ell^*\|_2,
\end{aligned}
$$

where the second inequality is due to the convexity of the surrogate loss $\ell$ such that $\langle \nabla L^\ell(w_{i-1}; D_i) - \nabla L^\ell(w_\ell^*; D_i), w_{i-1} - w_\ell^*\rangle \geq 0$. Thus we can see in both cases we have $\|w_i - w_\ell^*\|_2 \leq 2\|w_\ell^*\|_2$. Thus we complete the proof. $\square$

Next we will proof the main theorem.

Suppose there exists a $\tilde{t}$ such that for $i \le \tilde{t}$, we have $\|w_{i-1} - w_\ell^*\|_2 \ge \frac{\alpha}{3}$. Now consider in the $i$-th iteration where $i \le \tilde{t}$, if $\|w_{i-1} - w_\ell^*\|_2 \le \|w_\ell^*\|_2$. Then

$$
\begin{aligned}
\|w_i - w_\ell^*\|_2^2 &= \|w_{i-1} - w_\ell^*\|_2^2 - \eta \langle \nabla L^\ell(w_{i-1}; D_i) + \zeta_{i-1}, w_{i-1} - w_\ell^* \rangle + \eta^2 \|\nabla L^\ell(w_{i-1}; D_i) + \zeta_{i-1}\|_2^2 \\
&= \|w_{i-1} - w_\ell^*\|_2^2 - \eta \langle \nabla L^\ell(w_{i-1}; D_i) - \nabla L^\ell(w_\ell^*; D), w_{i-1} - w_\ell^* \rangle \\
&\quad + \eta^2 \|\nabla L^\ell(w_{i-1}; D_i) + \zeta_{i-1}\|_2^2 - \eta \langle \nabla L^\ell(w_\ell^*; D_i) + \zeta_{i-1}, w_{i-1} - w_\ell^* \rangle \\
&\le (1 - \tau\eta) \|w_{i-1} - w_\ell^*\|_2^2 - \eta\beta \|\nabla L^\ell(w_{i-1}; D_i) - \nabla L^\ell(w_\ell^*; D_i)\|_2^2 + 2\eta^2 \|\nabla L^\ell(w_{i-1}; D_i) - \nabla L^\ell(w_\ell^*; D_i)\|_2^2 \\
&\quad + 2\eta^2 \|\nabla L^\ell(w_\ell^*; D_i) + \zeta_{i-1}\|_2^2 - \eta \langle \nabla L^\ell(w_\ell^*; D_i) + \zeta_{i-1}, w_{i-1} - w_\ell^* \rangle \\
&\le (1 - \tau\eta) \|w_{i-1} - w_\ell^*\|_2^2 - \eta(\beta - 2\eta) \|\nabla L^\ell(w_{i-1}; D_i) - \nabla L^\ell(w_\ell^*; D_i)\|_2^2 + 4\eta^2 \|\nabla L^\ell(w_\ell^*; D_i)\|_2^2 + 4\eta^2 \|\zeta_{i-1}\|_2^2 \\
&\quad + \frac{\tau\eta}{2} \|w_{i-1} - w_\ell^*\|_2^2 + \frac{\eta}{\tau} \|\nabla L^\ell(w_\ell^*; D_i)\|_2^2 + \frac{\eta}{\tau} \|\zeta_{i-1}\|_2^2 \\
&\le (1 - \frac{\tau\eta}{2}) \|w_{i-1} - w_\ell^*\|_2^2 + \eta(4\eta + \frac{1}{\tau}) \|\nabla L^\ell(w_\ell^*; D_i)\|_2^2 + \eta(4\eta + \frac{1}{\tau}) \|\zeta_{i-1}\|_2^2,
\end{aligned}
$$

where the first inequality is due to Lemma 14. Since we have $\|w_{i-1} - w^*\|_2 \ge \frac{\alpha}{3}$, and take $\alpha = \sqrt{\frac{\tau^2}{9(4\tau\beta+1)}\alpha^2}$ in Lemma 13 and since $m \ge \tilde{\Omega}(\sqrt{d\frac{1}{\tau}(4\eta + \frac{1}{\tau})}\frac{(B+\sqrt{d}\|w_\ell^*\|_2)\sqrt{\log 1/\zeta \log 1/\delta}}{\epsilon\alpha})$ and $\|w_{i-1}\|_2 \le 3\|w_\ell^*\|_2$, we have for $i \le \tilde{t}$

$$
\|w_i - w_\ell^*\|_2^2 \le (1 - \frac{\tau\eta}{2})^i \|w_0 - w_\ell^*\|_2^2 + \frac{1}{\tau}(4\eta + \frac{1}{\tau})\frac{\tau^2}{9(4\tau\beta + 1)}\alpha^2 + \frac{\alpha^2}{9}
$$

$$
\le \|w_\ell^*\|_2^2 + \frac{2\alpha^2}{9} \le 4\|w_\ell^*\|_2^2.
$$

Thus, we can see that as long as for $i \le \tilde{t}$, $\|w_i - w^*\|_2 \ge \frac{\alpha}{3}$ and $\|w_0 - w_\ell^*\| \le \|w_\ell^*\|_2$ (this is true since $w_0 = 0$) we always have $\|w_i - w_\ell^*\|_2 \le 2\|w_\ell^*\|_2$. Thus, we can always use Lemma 14.

Now we consider several cases:

**Case 1:** If for all $i \le T$, $\|w_i - w_\ell^*\|_2 \ge \frac{\alpha}{3}$. Then by the above inequality we have

$$
\|w_T - w_\ell^*\|_2^2 \le (1 - \frac{\tau\eta}{2})^T \|w_0 - w_\ell^*\|_2^2 + \frac{1}{\tau}(4\eta + \frac{1}{\tau})\frac{\tau^2}{9(4\tau\beta + 1)}\alpha^2 + \frac{\alpha^2}{9} \le (1 - \frac{\tau\eta}{2})^T \|w_\ell^*\|_2^2 + \frac{2\alpha^2}{9}
$$

Thus, take $T = O(\frac{\log(\alpha/\|w_\ell^*\|_2)}{\log(1 - \frac{\tau\eta)}{2})}) = O(\frac{1}{\tau\eta}\log(\|w_\ell^*\|_2))$ we have

$$
\|w_T - w^*\|_2^2 \le \frac{2\alpha^2}{9} + \frac{2\alpha^2}{9} \le \frac{4\alpha^2}{9}. \tag{14}
$$

That is $\|w_T - w^*\|_2 \le \frac{2\alpha}{3}$.

**Case 2:** If Case 1 does not hold, then if there exist a $\tilde{t} < T$ (we assume $\tilde{t}$ is the largest one) such that when $i = \tilde{t}$ we have $\|w_i - w_\ell^*\|_2 \le \frac{\alpha}{3}$ and $\|w_i - w_\ell^*\|_2 \ge \frac{\alpha}{3}$ for $T \ge i \ge \tilde{t} + 1$. Then

$$
\begin{aligned}
\|w_{\tilde{t}+1} - w_\ell^*\|_2 &= \|w_{\tilde{t}} - w_\ell^* - \eta(\nabla L^\ell(w_{\tilde{t}}; D_i) + \zeta_{\tilde{t}})\|_2 \\
&\le \|w_{\tilde{t}} - w_\ell^* - \eta(\nabla L^\ell(w_{\tilde{t}}; D_i) - \nabla L^\ell(w_\ell^*; D_i))\|_2 + \eta\|\zeta_{\tilde{t}}\|_2 + \eta\|\nabla L^\ell(w_\ell^*; D_i)\|_2 \\
&\le \|w_{\tilde{t}} - w_\ell^*\|_2 + \eta\|\zeta_{\tilde{t}}\|_2 + \eta\|\nabla L^\ell(w_\ell^*; D_i)\|_2 \le \frac{\alpha}{3} + \frac{\alpha}{3} \le 2\|w_\ell^*\|_2,
\end{aligned}
$$

where the second inequality is due to the convexity of the surrogate loss $\ell$ such that $\langle \nabla L^\ell(w_{\tilde{t}}; D_i) - \nabla L^\ell(w_\ell^*; D_i), w_{\tilde{t}} - w_\ell^* \rangle \ge 0$. Thus, we can use the same argument as Case 1 and show that

$$
\|w_T - w_\ell^*\|_2^2 \le (1 - \frac{\eta\tau}{2})^{T-\tilde{t}-1} \|w_{\tilde{t}+1} - w_\ell^*\|_2^2 + \frac{2\alpha^2}{9} \le \frac{2}{3}\alpha^2. \tag{15}
$$

**Case 3** If Case 1 and Case 2 do not hold, then that is $\|w_T - w^*\|_2 \le \frac{\alpha}{3}$.

Thus, in total we must have $\|w_T - w_\ell^*\|_2 \le \alpha$ with probability at least $1 - 3\zeta$. Note that

$\|\nabla L_{\mathcal{P}}^\ell(w_T)\|_2 = \|\nabla L_{\mathcal{P}}^\ell(w_T) - \nabla L_{\mathcal{P}}^\ell(w_\ell^*)\|_2 \le \|w_T - w_\ell^*\|_2 \le \alpha$. Thus

$$L_{\mathcal{P}}(w) \le 2(1 + 2\mu)L_{\mathcal{P}}(w^*) + 4\mu\alpha.$$

Take $\alpha = \frac{\alpha}{4\mu}$ we can get the result. $\qquad\square$

## B   Omitted Proofs in Section 5

**Proof of Lemma 7.** We denote

$$\tilde{\ell}(w; x, y) = \int_0^{\langle w, \psi(x)\rangle + \phi(x)} (\sigma(z) - y)dz.$$

For any fixed $x$ we have

$$\mathbb{E}_y[\tilde{\ell}(w; x, y)] - \mathbb{E}_y[\tilde{\ell}(w^*; x, y)] = \mathbb{E}_y \int_{\langle w^*, \psi(x)\rangle + \phi(x)}^{\langle w, \psi(x)\rangle \phi(x)} (\sigma(z) - y)dz$$

$$= \int_{\langle w^*, \psi(x)\rangle + \phi(x)}^{\langle w, \psi(x)\rangle \phi(x)} (\sigma(z) - \mathbb{E}_y y)dz$$

$$= \int_{\langle w^*, \psi(x)\rangle + \phi(x)}^{\langle w, \psi(x)\rangle \phi(x)} (\sigma(z) - \sigma(\langle w^*, \psi(x)\rangle + \phi(x))))dz$$

$$= \int_{\langle w^*, \psi(x)\rangle + \phi(x)}^{\langle w, \psi(x)\rangle + \phi(x)} \frac{\sigma'(z)(\sigma(z) - \sigma(\langle w^*, \psi(x)\rangle + \phi(x)))}{\sigma'(z)}dz$$

$$\ge \frac{1}{2G}(\sigma(\langle w, \psi(x)\rangle + \phi(x)) - \sigma(\langle w^*, \psi(x)\rangle) + \phi(x))^2$$

$$\ge \frac{1}{2G}\left[\frac{(\sigma(\langle w, \psi(x)\rangle) - \sigma(\langle w^*, \psi(x)\rangle + \psi(x)))^2}{2} - (\sigma(\langle w, x\rangle + \psi(x)) - \sigma(\langle w, x\rangle))^2\right]$$

$$\ge \frac{1}{2G}\left[\frac{(\sigma(\langle w, \psi(x)\rangle) - \sigma(\langle w^*, \psi(x)\rangle + \psi(x)))^2}{2} - G^2 M^2\right].$$

On the other side we have $|\tilde{\ell}(w; x, y) - \ell(w; x, y)| = |\int_{\langle w, \psi(x)\rangle}^{\langle w, \psi(x)\rangle + \phi(x)}(\sigma(z) - y)dz| \le |\phi(x)| \le M$. In total take the expectation w.r.t $x$ we have

$$L_{\mathcal{P}}(w) - L_{\mathcal{P}}(w^*) \le 4G(L_{\mathcal{P}}^\ell(w) - L_{\mathcal{P}}^\ell(w^*)) + 2G^2 M^2 + 4GM.$$

$\qquad\square$

**Proof of Theorem 8.** It is sufficient for us to only consider the term of $L_{\mathcal{P}}^\ell(w) - L_{\mathcal{P}}^\ell(w^*)$. We can just use Theorem 1 to get the bound of $O(\frac{\sqrt{\theta \log \frac{1}{\delta}}}{n\epsilon} + \frac{1}{\sqrt{n}})$. For the other term, we can following the proof of Theorem 2. The only difference is that here we do not have (9) as $w^*$ is not the global minimizer of $L_{\mathcal{P}}^\ell(w)$. Thus, by the Lispchitz condition we have

$$\mathbb{E}_{\Phi, \tilde{D}}L^\ell(\Phi w^*, \tilde{D}) - L_{\mathcal{P}}^\ell(w^*) = \mathbb{E}_{\Phi, (x_i, y_i)\sim\mathcal{P}}[g^{y_i}(\langle\Phi w^*, \Phi x_i\rangle) - g^{y_i}(\langle w^*, x_i\rangle)]$$

$$\le 2\mathbb{E}_{\Phi, x}|\langle\Phi w^*, \Phi x\rangle - \langle w^*, x\rangle| = O(W\frac{\log n/\delta}{\sqrt{m}}).$$

Thus, similar to the proof of Theorem 2 in total we have

$$L_{\mathcal{P}}^\ell(w) - L_{\mathcal{P}}^\ell(w^*) \le \tilde{O}(W\frac{1}{\sqrt{m}} + W(\frac{\sqrt{m \log \frac{1}{\delta}}}{n\epsilon} + \frac{1}{\sqrt{n}})).$$

Take $m = O(\log(n/\delta)n\epsilon)$ we can get the result. $\qquad\square$

**Proof of Theorem 10 and 11 .** We first recall the following two lemmas that the neural networks we considered can be uniformly approximated.

**Lemma 16** (Goel & Klivans (2019)). *For $\mathcal{N}_2$ with the sigmoid function $\sigma_1$ and $G$-Lipschitz function $\sigma_2$, there exists a kernel $\mathcal{K}$ with $\mathcal{K}(x, x') \leq 1$ and feature map $\psi(x) \in \mathbb{R}^{D_m}$ ($D_m = 1 + d + \cdots + d^m$ and $m = O(\log(\frac{1}{\alpha_0}))$) such that $\mathcal{N}_2$ is $(\sqrt{k}\alpha_0, (\frac{\sqrt{k}}{\alpha_0})^C)$-uniformly approximated by kernel $\mathcal{K}$ with some constant $C > 0$ for any $\alpha_0$.*

**Lemma 17** (Goel & Klivans (2019)). *For $\mathcal{N}_2$ with the ReLU function $\sigma_1$ and $G$-Lipschitz function $\sigma_2$, there exists a kernel $\mathcal{K}$ with $\mathcal{K}(x, x') \leq 1$ and feature map $\psi(x) \in \mathbb{R}^{D_m}$ ($D_m = 1 + d + \cdots + d^m$ and $m = O(\frac{1}{\alpha_0})$) such that $\mathcal{N}_2$ is $(\sqrt{k}\alpha_0, 2^{C\frac{\sqrt{k}}{\alpha_0}})$-uniformly approximated by kernel $\mathcal{K}$ with some constant $C > 0$ for any $\alpha_0$.*

Thus combining with the previous two lemmas with $\alpha_0 = \frac{\alpha}{G\sqrt{k}}$ and Corollary 1 we have the proof. $\qquad\square$

## C   Omitted Proofs in Section 6

In this section we provide the proof of the theorem by applying the following technical lemmas. To begin with, we introduce some extra notations. Following Allen-Zhu et al. (2019), for a parameter collection $\mathbf{W}$ and $i \in [n]$, we denote the $l$-th hidden layer output of the network as

$$\mathbf{h}_{i,l} = \begin{cases} \sigma(\mathbf{W}_l \mathbf{h}_{i,l-1}) & \text{if } l \in [L-1] \\ \mathbf{x}_i & \text{if } l = 0 \end{cases}$$

We also define the binary diagonal matrices

$$\mathbf{D}_{i,l} = \text{diag}(\mathbb{I}\{(\mathbf{W}_l \mathbf{h}_{i,l})_1 > 0\}, ..., \mathbb{I}\{(\mathbf{W}_l \mathbf{h}_{i,l})_m > 0\}), l \in [L-1]$$

For $i \in [n]$ and $l \in [L-1]$, for the collection of initialization parameters $\mathbf{W}^{(0)}$, we use $\mathbf{h}_{i,l}^{(0)}, \mathbf{D}_{i,l}^{(0)}$ to denote the initial hidden layer outputs and binary diagonal matrices. We introduce the following matrix product notation used in the previous related work Zou et al. (2018); Cao & Gu (2019a;b):

$$\prod_{r=l_1}^{l_2} \mathbf{M}_r := \begin{cases} \mathbf{M}_{l_2} \mathbf{M}_{l_2-1} ... \mathbf{M}_{l_1} & \text{if } l_1 \leq l_2 \\ \mathbf{I} & \text{otherwise} \end{cases}$$

With this notation, we rewrite the neural network in the matrix representation from:

$$f(\mathbf{W}, \mathbf{x}_i) = \begin{cases} \sqrt{m} \cdot \mathbf{W}_L(\prod_{r=l+1}^{L-1} \mathbf{D}_{i,r}\mathbf{W}_r)\mathbf{h}_{i,l} & l \in [L-1] \\ \sqrt{m} \cdot \mathbf{W}_L \mathbf{h}_{i,l-1}^T & l = L \end{cases}$$

Under this notation, one can calculate the gradient of $f(\mathbf{W}, \mathbf{x}_i)$ as follows:

$$\nabla_{\mathbf{w}_l} f(\mathbf{W}, \mathbf{x}_i) = \begin{cases} \sqrt{m} \cdot [\mathbf{W}_L(\prod_{r=l+1}^{L-1} \mathbf{D}_{i,r}\mathbf{W}_r)\mathbf{D}_{i,l}]^\top \mathbf{h}_{i,l-1} & l \in [L-1] \\ \sqrt{m} \cdot \mathbf{h}_{i,l}^T & l = L \end{cases} \tag{16}$$

The following lemma shows the error between neural network function and its linearization under NTKF for all $\mathbf{W} \in \mathcal{B}(\mathbf{W}^{(0)}, \omega)$ with some small $\omega$.

**Lemma 18** (locally linearization of neural network, Lemma 4.1 in Cao & Gu (2019a)). *There exists an absolute constant $\kappa$ such that, with probability at least $1 - O(nL^2)\exp[-\Omega(m\omega^{2/3}L)]$ over the randomness of $\mathbf{W}^{(0)}$, for all $i \in [n]$ and $\mathbf{W} \in \mathcal{B}(\mathbf{W}^{(0)}, \omega)$ with $\omega \leq \kappa L^{-6}[\log(m)]^{-3/2}$*

$$|f(\mathbf{W}; \mathbf{x}_i) - f_{ntk}(\mathbf{W}; \mathbf{x}_i)| \leq O\left(\omega^{4/3} L^3 \sqrt{m\log(m)}\right)$$

Given the lemma 18, since the loss function is convex we can show the objective function is almost convex near the initialization. This implies the dynamics of the DP-SGD algorithm given in Algorithm 10 is similar to the dynamics of convex optimization.

**Lemma 19** (locally almost convexity). *There exists an absolute constant $\kappa$ such that, with probability at least $1 - O(nL^2)\exp[-\Omega(m\omega^{2/3}L]$ over the randomness of $\mathbf{W}^{(0)}$, for all $i \in [n]$ and $\mathbf{W}', \mathbf{W} \in \mathcal{B}(\mathbf{W}^{(0)}, \omega)$, any $\Delta > 0$, with $\omega \leq \kappa L^{-6}m^{-3/8}[\log(m)]^{-3/2}\Delta^{3/4}$ it holds uniformly*

$$L_i(\mathbf{W}') \geq L_i(\mathbf{W}) + \langle \nabla L_i(\mathbf{W}), \mathbf{W}' - \mathbf{W} \rangle - \Delta$$

**Proof of Lemma 19.** By the convexity of loss function $\ell$ we have

$$\begin{aligned} L_i(\mathbf{W}') - L_i(\mathbf{W}) &= \ell(f(\mathbf{W}'; \mathbf{x}_i), y_i) - \ell(f(\mathbf{W}; \mathbf{x}_i), y_i) \\ &\geq \ell'(f(\mathbf{W}; \mathbf{x}_i), y_i) \cdot (f(\mathbf{W}'; \mathbf{x}_i) - f(\mathbf{W}; \mathbf{x}_i)). \end{aligned}$$

By the triangular inequality

$$\begin{aligned} |\ell'(f(\mathbf{W}; \mathbf{x}_i), y_i) \cdot (f(\mathbf{W}'; \mathbf{x}_i) - f(\mathbf{W}; \mathbf{x}_i))| &\geq |\ell'(f(\mathbf{W}; \mathbf{x}_i), y_i) \cdot \langle \nabla f(\mathbf{W}; \mathbf{x}_i), \mathbf{W}' - \mathbf{W} \rangle| \\ &\quad - |\ell'(f(\mathbf{W}; \mathbf{x}_i), y_i) \cdot (f(\mathbf{W}'; \mathbf{x}_i) - f(\mathbf{W}; \mathbf{x}_i) - \langle \nabla f(\mathbf{W}; \mathbf{x}_i), \mathbf{W}' - \mathbf{W} \rangle)| \end{aligned}$$

where we could decompose the last term in the right side of inequality into $|\ell'(f(\mathbf{W}; \mathbf{x}_i), y_i)| \cdot ([f(\mathbf{W}'; \mathbf{x}_i) - f_{ntk}(\mathbf{W}'; \mathbf{x}_i)] - [f(\mathbf{W}; \mathbf{x}_i) - f_{ntk}(\mathbf{W}; \mathbf{x}_i)])$. Now we can apply the linearization approximation Lemma 18 and $\ell(\cdot)$ is $S$-lipschitz with respect to $\mathbf{W}$ to obtain the following inequality

$$\begin{aligned} L_i(\mathbf{W}') - L_i(\mathbf{W}) &\geq \langle \nabla L_i(\mathbf{W}), \mathbf{W}' - \mathbf{W} \rangle - O(S\omega^{4/3}L^3\sqrt{m\log(m)}) \\ &\geq \langle \nabla L_i(\mathbf{W}), \mathbf{W}' - \mathbf{W} \rangle - \Delta \end{aligned}$$

The last inequality holds if $\omega \leq \kappa L^{-6}m^{-3/8}[\log(m)]^{-3/2}\Delta^{3/4}$ for some constant $\kappa$. $\qquad\square$

With the lemma 19, it is clear the loss of neural network is almost convex. This inspired us to analyze the dynamics of the DP-SGD algorithm 10. By carefully selecting the learning rate and a number of iterations, **the DP-SGD algorithm is similar to the noised SGD convex optimization**. Algorithm 5 is similar to the dynamics of convex optimization. In the following, we will show the loss function is locally Lipschitz.

**Lemma 20** (Lemma 7.1 in Allen-Zhu Allen-Zhu et al. (2019)). *If $\epsilon \in (0, 1]$, with probability at least $1 - O(nl) \cdot e^{\Omega(m\epsilon^2/L)}$ over the randomness of $\mathbf{W}^{(0)}$, we have*

$$\forall i \in [n], l \in [L] : ||h_{i,l}|| \in [1 - \epsilon, 1 + \epsilon] \tag{17}$$

**Lemma 21** (Lemma 8.2 in Allen-Zhu Allen-Zhu et al. (2019)). *Suppose $\omega \leq \frac{1}{CL^{9/2}\log^3 m}$ for some sufficiently large constant $C > 1$. With probability at least $1 - e^{-\Omega(m\omega^{2/3}L)}$, for every $\mathbf{W} \in B(\mathbf{W}^{(0)}, \omega)$,*

$$||h_{i,j} - h_{i,j}^{(0)}|| \leq O(\omega L^{5/2}\sqrt{\log m}) \tag{18}$$

**Lemma 22** (Locally Bounded Gradient). *There exists an absolute constant $\kappa$ such that, with probability at least $1 - O(nL)\exp[-\Omega(m\omega^{2/3}L]$ over the randomness of $\mathbf{W}^{(0)}$, for all $i \in [n]$, $l \in [L]$ and $\mathbf{W} \in \mathcal{B}(\mathbf{W}^{(0)}, \omega)$, with $\omega = \frac{R}{\sqrt{m}} \leq \kappa L^{-6}[\log m]^{-3}$*

1. $||\nabla_{\mathbf{W}_l} f(\mathbf{W}; \mathbf{x}_i)||_F = O(\sqrt{m})$

2. $||\nabla_{\mathbf{W}_l} L_i(\mathbf{W})||_F = O(S\sqrt{m})$

**Proof of Lemma 22.** Observing that the loss function $\ell(\cdot, y_i)$ is assumed to be $S$-lipschitz for any $y_i$, it is sufficient to show that the gradient of $f_{ntk}(\mathbf{W}, x)$ is bound with high probability.
By Lemma 20, with probability at least $1 - O(nL) \cdot \exp[-\Omega(m/L)]$, $||\mathbf{h}_{i,l}^0||_2 \in [3/4, 5/4]$ for all $i \in [n]$ and $l \in [L - 1]$. Moreover, by Lemma 21 and the fact that $\sigma(\cdot)$ is of 1-lipschitz continuity, with probability $1 - O(nL) \cdot \exp[-\Omega(m\omega^{2/3}L]$, $||\mathbf{h}_{i,l} - \mathbf{h}_{i,l}^{(0)}||_2 \leq O(\omega L^{5/2}\sqrt{\log m})$. Therefore, by the setting of neighborhood

$\omega = R \cdot m^{-1/2}$ and the assumption of $m$, we have $||\mathbf{h}_{i,l}||_2 \in [1/2, 3/2]$ for all $i \in [n]$ and $l \in [L-1]$. Note by Lemma 19 that this implicitly indicates that

$$\frac{R}{\sqrt{m}} \leq \kappa L^{-6} m^{-3/8} [\log(m)]^{-3/2} \Delta^{3/4} \implies m^{\frac{1}{8}} \geq \tilde{\Omega}(RS^{\frac{3}{4}} L^{\frac{9}{4}} \Delta^{\frac{3}{4}}). \tag{19}$$

The above statement tells that the output of arbitrary hidden-layer $\mathbf{h}_{i,l}$ lies in a small region, therefore plugging in (16) for $\nabla_{\mathbf{W}_l} f(\mathbf{W}; \mathbf{x}_i)$ we can get the desired result. $\qquad \square$

s

**Lemma 23.** *When $M \geq \Omega(\log \frac{T}{\gamma})$ we have with probability at least $1 - \gamma$ for all $t \in [T]$, $|B_t| \geq C_1 M$ for some constant $C_1 > 0$*

*Proof.* By the subsampling procedure we can easily see $\mathbb{E}[|B_t|] = qn = M$. Thus, by the Multiplicative Chernoff bound we can see for all $t \in [T]$

$$\mathbb{P}(||B_t| - M| \geq \gamma M) \leq 2 \exp(-\frac{\gamma^2 M}{3})$$

Thus, we have with probability at least $1 - \gamma$ we have $|B_t| \geq (1 - \frac{\sqrt{3 \log \frac{T}{\gamma}}}{\sqrt{M}})M$. Thus when $M \geq \Omega(\log \frac{T}{\gamma})$ we have the result. $\qquad \square$

**Lemma 24.** *Gaussian vector norm tail bound Let $\mathbf{X} \sim N(\mu, \sigma^2 I)$ where $\mu \in \mathbb{R}^n$ and $\sigma \in \mathbb{R}$. For any $t > 0$, with probability at most $1 - 2\exp(-\frac{t}{2n\sigma^2})$*

$$||\mathbf{X} - \mu||_F \leq t$$

**Proof of the Lemma 24.** Let $\mathbf{Y} \sim N(0, I)$, then $||\mathbf{X} - \mu||_F =^d ||\sigma \mathbf{Y}||_F$. For all $t > 0$ and $s > 0$, based on the inequality from Lemma 4 in Rhee & Talagrand (1986), we have

$$P(||\sigma \mathbf{Y}||_F > t) \leq P(||\sigma \mathbf{Y}||_1 > t)$$

$$\leq e^{-st} \prod_{i=1}^{n} \mathbb{E}[\exp(t\sigma |\mathbf{Y}_i|)]$$

$$\leq 2 \exp(s^2 n\sigma/2 - st)$$

$$\leq \min_s 2 \exp(s^2 n\sigma/2 - st)$$

$$\leq 2 \exp(\frac{t^2}{2n\sigma})$$

$\qquad \square$

**Lemma 25** (Dynamically Cumulative Loss). *If Lemma 23 holds, $C \leq O(\min\{SL\sqrt{m}, R\})$ and $n \geq \tilde{\Omega}(\frac{C(\sqrt{L}m + \sqrt{m}d)\sqrt{T \log(1/\gamma) \log(1/\delta)}}{R\epsilon})$, then with probability at least $1 - O(nL^2)\exp[-\Omega(m\omega^{2/3}L] - \gamma$ over the randomness of $\mathbf{W}^{(0)}$ and the noise, for all $t \in [T]$ and $\mathbf{W}^* \in \mathcal{B}(\mathbf{W}^{(0)}, R/\sqrt{m})$, any $\Delta > 0$, with set size $\eta T = \Theta(\frac{SL^2 R^2}{\kappa C \sqrt{m} \Delta}), \Delta = O(\frac{SL^{\frac{3}{2}} R}{\sqrt{T}})$ with $m \geq O(L^{56} R^{24} \Delta^{-14} S^{-8} C^{-8} [\log(m)]^{12})$ it holds uniformly*

$$\sum_{t=1}^{T} L_t(\mathbf{W}^{(t)}) - L_i(\mathbf{W}^*) \leq \frac{SL\eta\sqrt{m}}{2\kappa C} \sum_{t=1}^{T} ||\mathbf{G}_t||_F^2 + 3T\Delta$$

**Proof of Lemma 25.** First we show that following the DP-SGD update rule, the parameters would be restricted within a small region near to initialization by choosing some artificial parameters. By Lemma 18,

19, 22 there exists some small enough positive constant $C_1$, suppose $\omega = C_1 \cdot L^{-6} m^{-3/8} [\log(m)]^{-3/2} \Delta^{3/4} \geq \frac{R}{\sqrt{m}}$ such that the conditions in the above mentioned three lemma hold. Recall the update rule is

$$\mathbf{W^{(t+1)}} \leftarrow Proj_{\mathcal{W}}(\mathbf{W}^{(t)} - \eta \cdot (\frac{1}{|B_t|} \sum_{(x_j^{(t)}, y_j^{(t)}) \in B_t} \tilde{g}_t(x_j^{(t)}) + \mathbf{G}_t)).$$

Assume the sample size $n \geq \tilde{\Omega}(\frac{C(\sqrt{L}m + \sqrt{md})\sqrt{T \log(1/\gamma) \log(1/\delta)}}{R\epsilon})$ , we can have the following inequality with probability at least $1 - \gamma$

$$
\begin{aligned}
||\mathbf{W}_l^{(T)} - \mathbf{W}_l^{(0)}||_F &\leq \sum_{t=1}^{T} ||\mathbf{W}_l^{(t)} - \mathbf{W}_l^{(t-1)}||_F \\
&\leq \eta \sum_{t=1}^{T} ||\frac{1}{|B_t|} \sum_{(x_j^{(t)}, y_j^{(t)}) \in B_t} \tilde{g}_t(x_j^{(t)}) + \mathbf{G}_t||_F \\
&\leq 2T\eta R \\
&\leq O(\frac{SL^2 R^3}{\Delta C \sqrt{m}}) \\
&\leq \omega
\end{aligned}
$$

The first inequality follows by the triangle inequality. The second inequality could be seen from the update rule. The third inequality holds since by Lemma 23 and Gaussian tail bound we have with probability at least $1 - \gamma$, for all $t \in [T]$ both $|B_t| \geq \Omega(M)$ and $||G_t||_F \leq \tilde{O}(\frac{M(\sqrt{L}m + \sqrt{md})C\sqrt{T \log 1/\delta \log 2T/\gamma}}{n|B_t|\epsilon})$ holds, which indicate $\sum_{t=1}^{T} ||\mathbf{G}_t||_F \leq TR$ if $n \geq \tilde{\Omega}(\frac{C(\sqrt{L}m + \sqrt{md})\sqrt{T \log(2/\gamma) \log(1/\delta)}}{R\epsilon})$, and the assumption that $C \leq R$. The fourth inequality holds due to that

$$T\eta \leq O(\frac{SL^2 R^2}{\Delta C \sqrt{m}}). \tag{20}$$

The last inequality holds, if $m \geq \Omega \left( S^{-8} C^{-8} L^{56} R^{24} \Delta^{-14} [\log(m)]^{12} \right)$. Thus we have $\mathbf{W}^t \in \mathcal{B}(\mathbf{W}^{(0)}, w)$ with high probability for all $t \in [T]$. Suppose $\mathbf{W}^* \in \mathcal{B}(\mathbf{W}^{(0)}, \omega^* = R/\sqrt{m})$. Following above setting, Lemma 18, 19, 22 hold. Then for any positive constant $\Delta > 0$ the following inequality holds.

$$
\begin{aligned}
L_t(\mathbf{W}^{(t)}) - L_t(\mathbf{W}^*) &\leq \langle \nabla_{\mathbf{W}} L_t(\mathbf{W}^{(t)}), \mathbf{W}^{(t)} - \mathbf{W}^* \rangle + \Delta \\
&= \frac{1}{\eta} \max(1, \frac{||\mathbf{g}_t(x_t)||_F}{C}) \langle \eta \tilde{\mathbf{g}}_t(x_t), \mathbf{W}^{(t)} - \mathbf{W}^* \rangle + \Delta \\
&= \frac{1}{2\eta} \max(1, \frac{||\mathbf{g}_t(x_t)||_F}{C})(\eta^2 C^2 + ||\mathbf{W}^{(t)} - \mathbf{W}^*||_F^2 \\
&\quad - ||\mathbf{W}^{(t)} - \mathbf{W}^* - \eta \tilde{\mathbf{g}}_t(x_t)||_F^2) + \Delta \\
&\leq \max(1, \frac{||\mathbf{g}_t(x_t)||_F}{C})\{\frac{\eta C^2}{2} + \frac{1}{2\eta}[||\mathbf{W}^{(i)} - \mathbf{W}^*||_F^2 \\
&\quad - ||\mathbf{W}^{(i+1)} - \mathbf{W}^*||_F^2] + \frac{\eta}{2}||\mathbf{G}_i||_F^2 \\
&\quad + \frac{\eta}{2M} \sum_{j=1}^{M} ||\tilde{\mathbf{g}}_t(x_j) - \tilde{\mathbf{g}}_t(x_t)||_F^2\} + \Delta \\
&\leq \max(1, \frac{||\mathbf{g}_t(x_t)||_F}{C})\{\frac{3\eta C^2}{2} + \frac{1}{\eta}[||\mathbf{W}^{(t)} - \mathbf{W}^*||_F^2 \\
&\quad - ||\mathbf{W}^{(t+1)} - \mathbf{W}^*||_F^2] + \frac{\eta}{2}||\mathbf{G}_t||_F^2\} + \Delta
\end{aligned}
$$

The first inequality follows by lemma 19. The second equality is a direct application from the definition of inner product in metric space. The third equality holds because the connection between inner product and

norm in metric space.

Thus by telescope summation and simply removing the negative term, the cumulative loss could be bounded by any loss near the initialization parameters

$$\sum_{t=1}^{T} L_t(\mathbf{W}^{(t)}) \leq \sum_{t=1}^{T} L_t(\mathbf{W}^*) + \max_{t \in [T]}(1, \frac{||\mathbf{g}_t(x_t)||_F}{C})\{\frac{3T\eta C^2}{2} + \frac{1}{\eta}\sum_{t=1}^{T}[||\mathbf{W}^{(t)} - \mathbf{W}^*||_F^2$$

$$- ||\mathbf{W}^{(t+1)} - \mathbf{W}^*||_F^2] + \frac{\eta}{2}\sum_{t=1}^{T}||\mathbf{G}_t||_F^2\} + T\Delta$$

$$= \sum_{t=1}^{T} L_t(\mathbf{W}^*) + \max_{t \in [T]}(1, \frac{||\mathbf{g}_t(x_t)||_F}{C})\{\frac{3T\eta C^2}{2} + \frac{1}{\eta}[||\mathbf{W}^{(0)} - \mathbf{W}^*||_F^2$$

$$- ||\mathbf{W}^{(T)} - \mathbf{W}^*||_F^2] + \frac{\eta}{2}\sum_{t=1}^{T}||\mathbf{G}_t||_F^2\} + T\Delta$$

$$\leq \sum_{t=1}^{T} L_i(\mathbf{W}^*) + O(\frac{SL\sqrt{m}}{C}\{\frac{3T\eta C^2}{2} + \frac{LR^2}{2\eta m} + \frac{\eta}{2}\sum_{t=1}^{T}||\mathbf{G}_t||_F^2\} + T\Delta)$$

$$\leq \sum_{t=1}^{T} L_t(\mathbf{W}^*) + O(\frac{SL\eta\sqrt{m}}{C}\sum_{t=1}^{T}||\mathbf{G}_t||_F^2 + T\Delta)$$

The third inequality holds because $C \leq \max_{t \in [T]} ||\mathbf{g_t}(\mathbf{x_t})||_F = O(SL\sqrt{m})$. The last inequality hold if

$$SL\sqrt{m}T\eta C \leq O(T\Delta) \implies \eta \leq O(\frac{\Delta\kappa}{SL\sqrt{m}C}) \tag{21}$$

$$\frac{SL^2R^2}{C\eta\sqrt{m}} \leq O(T\Delta) \implies \eta T \geq \Omega(\frac{SL^2R^2}{\kappa C\sqrt{m}\Delta}) \tag{22}$$

Thus, combining (20), (21) and (22) we must have

$$\tag{23}$$

$$\Delta \geq \Omega(\frac{SL^{\frac{3}{2}}R}{\sqrt{T}}). \tag{24}$$

$\square$

Next we can verify the loss function $L_i(\mathbf{W}^{(i)})$ in the DP-SGD algorithm 10 is bounded from above. This could be seen from the Gaussian tail bound for the initial state network, and the locally stability of the loss function.

**Lemma 26** (Lemma 4.4 in Cao & Gu (2019a)). *With probability at least $1 - \xi$, for all $i \in [n]$ with $\omega \leq L\log(nL/\xi)$, we have*

$$||f_{\mathbf{W}^{(0)}}(\mathbf{x}_i)||_F \leq O(\sqrt{\log(n/\xi)})$$

**Lemma 27** (Stability of the Loss function). *There exists an absolute constant $\kappa$ such that, with probability at least $1 - \xi$, for all $\mathbf{W} \in \mathcal{B}(\mathbf{W}^{(0)}, R/\sqrt{m})$, with $m \geq O(R^{-1}L^{-3/2}[\log(nL/\xi)]^3)$*

$$L_i(\mathbf{W}) - L_i(\mathbf{W}^{(0)}) \leq O(SLR)$$

**Proof of the Lemma 27.** Since $\ell(\cdot)$ is an Lipschitz function and suppose Lemma22 condition holds, then with probability at least $1 - O(nL^2)\exp[-\Omega(m\omega^{2/3}L]$, the inequality below holds

$$L_i(\mathbf{W}) - L_i(\mathbf{W}^{(0)}) \leq S \cdot \langle f'(\mathbf{W}), \mathbf{W} - \mathbf{W}^{(0)}\rangle$$

$$\leq S \cdot \sum_{l=1}^{L} \langle \nabla_{\mathbf{W}_l} f(\mathbf{W}), \mathbf{W} - \mathbf{W}^{(0)}\rangle$$

$$\leq O(SLR)$$

The probability could be reduced to $1-\xi$ if $m \geq O(R^{-1}L^{-3/2}[\log(nL/\xi)]^3)$, we obtain the desired result. $\square$

*Remark* 17. Combine Lemma 26 and the Lemma 27, with probability at least $1-2\xi$, we immediately obtain the upper bound for the empirical loss function, suppose $\omega = R/\sqrt{m}$

$$L_i(\mathbf{W}) \leq O(\sqrt{\log(2n/\xi)} + SLR)$$

The probability could be reduced to $1-\xi$ by normalize the coefficients with some constants.

**Proof of the Theorem 13** . By Lemma 25, 27, converting the condition of Lemma 25 with respect to $m$, with probability at least $1-\xi-\gamma$ and there exists $m \geq \tilde{\Omega}(L^{56}R^{16}\Delta^{-14}S^{-2}C^{-8}[\log(nL+1/\xi)]^3)$, $n \geq \tilde{\Omega}(\frac{C(\sqrt{L}m+\sqrt{m}d)\sqrt{T\log(1/\gamma)\log(1/\delta)}}{R\epsilon})$ such that all lemmas hold, therefore we can apply the Azuma-Hoeffding inequality or so called online-to-batch technique to get the expectation loss

$$\frac{1}{T}\sum_{t=1}^{T}L_{\mathcal{D}}(\mathbf{W}^{(t)}) \leq \frac{1}{T}\sum_{t=1}^{T}L_t(\mathbf{W}^{(t)}) + I \cdot \sqrt{\frac{2\log(1/\xi)}{T}}$$

$$\leq \frac{1}{T}\sum_{t=1}^{T}L_t(\mathbf{W}^*) + O(\frac{SL\eta\sqrt{m}}{2CT}\sum_{t=1}^{T}\|\mathbf{G}_t\|_F^2 + \Delta + I \cdot \sqrt{\frac{\log(1/\xi)}{T}})$$

The second inequality holds because of Lemma 25, and the upper bound of Loss function in remark 17 $I = O(\sqrt{\log(2n/\xi)} + \frac{SLR}{\sqrt{m}})$ with probability $1-3\xi$. Thus under the previous lemma we have with probability $1-\gamma-4\xi$

$$\frac{1}{T}\sum_{t=1}^{T}L_{\mathcal{D}}(\mathbf{W}^{(t)}) \leq \frac{1}{T}\sum_{t=1}^{T}L_t(\mathbf{W}^*) + O(\frac{SL\sqrt{m}\eta\sigma^2(m^2L+md)}{C} + \Delta + I \cdot \sqrt{\frac{\log(1/\xi)}{T}})$$

$$\leq \frac{1}{T}\sum_{t=1}^{T}L_t(\mathbf{W}^*) + \tilde{O}(\frac{SL^{3/2}R\sqrt{T}\log(1/\delta)\log(1/\gamma)m^2}{n^2\epsilon^2}(L+d/m) + \frac{SL^{\frac{3}{2}}R}{\sqrt{T}}$$

$$+ I\sqrt{\frac{\log(1/\xi)}{T}})$$

where $\eta = \Theta(\frac{SL^2R^2}{CT\sqrt{m}\Delta})$, $\Delta = O(\frac{SL^{\frac{3}{2}}R}{\sqrt{T}})$, $\sigma^2 = \tilde{O}(\frac{TC^2\log(1/\delta)}{n^2\epsilon^2})$ for some small enough constant $\kappa \geq 0$. Since $\mathbf{W}^* \in \mathcal{B}(\mathbf{W}^{(0)}, \omega = R/\sqrt{m})$, by Lemma 18, with probability at least $1-\xi$

$$L_i(\mathbf{W}^*) = \ell(f(\mathbf{W}^*; \mathbf{x}_i))$$

$$\leq \ell(f_{ntk}(\mathbf{W}^*; \mathbf{x}_i)) + O(S\omega^{4/3}L^3\sqrt{m\log(m)})$$

$$= \ell(f_{ntk}(\mathbf{W}^*; \mathbf{x}_i)) + O(SR^{4/3}L^3\sqrt{\log(m)}m^{-1/6})$$

$$\leq \ell(f_{ntk}(\mathbf{W}^*; \mathbf{x}_i)) + O(\frac{SLR}{\sqrt{T}})$$

The first inequality results from the $S$-Lipschitz continuity of $\ell(\cdot)$ and Lemma18. And we can simplify the left term assuming $m \geq \Omega(\log(m)^3L^{12}R^2T^3)$. Since this holds for any $\mathbf{W}^* \in \mathcal{B}(\mathbf{W}^{(0)}, R/\sqrt{m})$ we can take the infimum over it, plugging into the above bound

$$\mathbb{E}_{\mathcal{A}_{SGD}}[\frac{1}{T}\sum_{i=1}^{T}L_{\mathcal{D}}(\mathbf{W}^{(i)})] \leq \inf_{f \in \mathcal{F}(\mathbf{W}^{(0)}, R/\sqrt{m})}\{\frac{1}{T}\sum_{i=1}^{T}\ell(f(\mathbf{x}_i))\} + \tilde{O}(\frac{SL^{3/2}R\sqrt{T}\log(1/\delta)m^2}{n^2\epsilon^2}(L+d/m)$$

$$+ \frac{SL^{\frac{3}{2}}R}{\sqrt{T}} + I\sqrt{\frac{\log(1/\xi)}{T}})$$

$$\leq \inf_{f \in \mathcal{F}(\mathbf{W}^{(0)}, R/\sqrt{m})}\{\frac{1}{T}\sum_{i=1}^{T}\ell(f(\mathbf{x}_i))\} + SL^{\frac{3}{2}}R \cdot \tilde{O}(\frac{\max(L, \frac{d}{m})\log(1/\delta)m^2\sqrt{T}}{n^2\epsilon^2}$$

$$+ \frac{1}{\sqrt{T}} + \sqrt{\frac{\log(1/\xi)}{T}})$$

The inequalities hold by plug in the $I$ and ignore the logarithmic term with respect to $n$ for conciseness purpose.

The Lemma 19,22 reveal both the local landscape and training dynamic of DP-NN. Following the similar procedure, we can rewrite the almost convexity in the summation form: $L_\mathcal{D}(\hat{\mathbf{W}}) = L_\mathcal{D}(\frac{1}{T}\sum_{i=1}^{T}\mathbf{W}^{(i)}) \leq \frac{1}{T}\sum_{i=1}^{T}L_\mathcal{D}(\mathbf{W}^{(i)}) + \Delta$, plugging in yields the desired result. Finally, note that in the previous lemma we need

$$m \geq O(L^{56}R^{24}\Delta^{-14}S^{-8}C^{-8}[\log(m)]^{12}), \tag{25}$$

plugging $\Delta$ and by the non-negativity of the loss function we can get the result. $\qquad\square$

