# OpenReview forum: "Differentially Private Non-convex Learning for Multi-layer Neural Networks"
_TMLR — Withdrawn by Authors_

### Review · Reviewer_YPJp · 2024-06-13

**Summary Of Contributions:**

This paper considers differentially private neural network models and attempts to bound the population risk on learnt models.

The contributions are not very clear to me.  First, the paper does not clearly describe how the results go beyond the related work.  Second, given the current text I'm not convinced the claims are correct.

**Audience:**

Yes

**Claims And Evidence:**

No

**Requested Changes:**

* First, it is important to make the text much more rigorous so it becomes possible to verify the correctness of the claims.
* Second, please check the correctness of the claims, my strong feeling currently is that at least some of the theorems are not correct in their current form.

**Strengths And Weaknesses:**

### main points

The paper studies an interesting problem.

The presentation can be substantially improved.  The paper contains many unclear or inaccurate or even incorrect sentences.  A small sample is given in "details" below.

Some of the results seem flawed.  For example, I fail to understand the proof for Theorem 1.

* Page 22 says that $w_\ell^* = argmin_w L_\mathcal{P}^\ell(w)$.
* Later, the text says that according to Lemma 3, $w_* = W_*^\ell$.
* However, the statement of lemma 3 only contains $w_*$, it does not contain $w_*^\ell$ so it is unclear how it can say something about the relation between the two variables.  In fact, changing the loss function or the gradient function in some other scenarios does change the position of the optimum.  It would surprise me the optimum is not affected by replacing the non-convex loss by a convex loss.

### Details

* Sec 1: "Firstly, most of the existing work adopts the gradient norm
of the population risk function to measure the utility," : this sentence is unclear.  Usually one measures the utility with metrics such as accuracy, MSE, AuROC, ... or the excess population risk as you suggest for the convex case.

* "most research has narrowly focused on general non-convex loss functions, overlooking the intricacies of neural network structures." : this sentence too is rather mysterious.  Neural networks are very flexible models, it is known that sufficiently large neural networks can approximate any continuous function (satisfying some reasonable conditions), similarly it seems that using a few constant inputs one can approximate any continuous loss function with a neural network loss function (under some reasonable conditions), so from the statement in the paper only it is hard to see how either (a) one can exploit the knowledge that we are dealing with a neural network loss function to make the problem easier, or (b) in what sense you want to consider  a neural network loss function as harder than the general case.

* "we introduce an $(\epsilon, \delta)$-DP algorithm and demonstrate its efficacy with an output upper bound" -> Are you sure "output bound" is correct?  Usually one bounds the error or other property of the output, but bounding the output would mean that for large datasets the model always predicts a value very close to 0.

* It seems that what in the paper is called a link function may be the same as what in the neural network community is usually called "activation function".  Please clarify the terminology to the reader.

* "In this scenario, we establish that an upper bound of ... is feasible." : here too, please specify clearly what you upper bound.

* "our attention pivots to the misspecified mode" : please define "misspecified model" (and if possible provide a citation), it doesn't seem a very well-known concept to me.

* "Jain & Thakurta (2014) provided the first study on DP-GLL and showed that in the unconstrained case, the error bound can achieve ... in general," : do you mean that the authors prove this bound?  Or that in worst case the error can be that large (a kind of lower bound) ?  The grammar is not fully clear.

* Section 2 describes some related work but does not do much effort to relate this work to the current paper, i.e., Sec 1 nor Sec 2 explain how the current paper goes beyond the state of the art and is novel.  While novelty is only a criterion for ICLR (and not so much for TMLR), TMLR requires that the paper describes clear claims (even in the case where they are not so novel).

* Sec 3: "Differentially Private Stochastic Optimization (DP-SO) is to find a model $w^{priv}$ to minimize the population risk, i.e., ... with the guarantee of being differentially private" : In fact, $w^{priv}$  does not minimize the population risk (can not due to the privacy requirements), so it would be better to say "approximately".  Please notice that it would also be wrong to say things like "find a model $w^{priv}$ in expectation minimizing the population risk subject to being differentially private", since the Abadi paper does not prove such optimality.

* "where the expectation takes over the randomness" -> "where the expectation is taken over the randomness"

* "The utility of $w^{priv}$ is measured by the (expected) excess population risk" : as far as I understand (and please define more precisely the used terminology if my understanding is incorrect) $w^{priv}$ can't be measured in this way because one typically doesn't have access to the full population (only to a sample, allowing to

* Def 6: "A random matrix satisfies Johnson-Lindentrauss (JL) property if ..." : "property" needs an article, e.g., "the Johnson-Lindentrauss (JL) property".

* Def 6: the notation $\Phi_u$ is not defined.  The vector norm $\\|u\\|_2$ suggests that $u$ is a vector, but it also occurs as the index of a matrix.  Moreover, $<\Phi_u, \Phi_v>$ suggests that $\Phi_u$ is not a scalar, so $u$ is not a pair of indices.  If $u$ would be a single integer, it would be unclear whether $\Phi_u$ is the $u$-th column or the $u$-th row of $\Phi$.

* "Specifically, when R is a random Gaussian matrix with ..." : $\alpha$ and $\beta$ are not parameters of a random Gaussian matrix.  Random Gaussian matrices are commonly characterized by their shape (in this case $k\times d$) and the mean and variance of every cell of the matrix (but there is no information about $\alpha$ or $\beta$ being the mean or the variance).  Even if $\alpha$ or $\beta$ would denote the mean or the variance of individual matrix cells, constraining $k$ doesn't seem to make $(\alpha,\beta)$ equal to $(0,1)$.

* Sec 4: I guess (based on the intro to Sec 4) that "well-specified" here does not mean that something is well specified but simply that the optimal model is in the hypothesis space.
* Eq (1) suggests that a link function is an activation function.  Most popular activation functions (sigmoid, relu, ...) are still quasi-convex, but the text doesn't seem to investigate whether such partial properties can still be exploited (in at least some cases they can).

* Assumption 1 : please define "non-monotone decreasing".  Do you mean it is not (monotonically decreasing), Or do you mean it is decreasing but not monotonically ?  The former is a very weak property, e.g., the function could be monotonically decreasing everywhere except for an arbitrarily small set where it is increasing.  The latter would be a rather unclear concept.

* "to approximate the population function"  : as an abbreviation of "population risk function", the term "population risk" is more precise than "population function", since there may exist functions on the population which are not risks.

* Eq (2) : even if others have considered this, the paper doesn't motivate Eq (2) in the context of the current paper.  The fact that a non-convex loss function such as the sigmoid is chosen in usually intentional, e.g., because one wants a bounded loss function.  If an unbounded or even convex loss function would have been appropriate, the designer of the application would have started with such a convex loss function i nthe first place.  Years of research have been spent on being able to work with non-convex loss functions (e.g., logistic regression) and have shown that in some applications these give better results. Notice that in such existing research, one aims at finding the optimum despite the non-convex loss function.  So replacing it with a convex surrogate would deserve some motivation at the start.

* The text around Theorem 1 nor Theorem 2 points to where there is a proof for these theorems.

The remainder of the text too contains many similarly unclear sentences and minor mistakes.

---

### Review · Reviewer_Ao6E · 2024-06-29

**Summary Of Contributions:**

The paper addresses differentially private stochastic optimization for fully connected neural networks with a single output node. It starts with generalized linear models (GLMs) without hidden nodes, focusing on cases with zero-mean noise and bounded, Lipschitz continuous link functions. The authors propose several algorithms. They also explore scenarios with ReLU link functions, and compare well-specified and misspecified models using ReLU regression as an example. The paper then extends these ideas to two-layer neural networks with sigmoid or ReLU activation functions in well-specified models. In the third part, it studies the theoretical guarantees of DP-SGD (Abadi et al., 2016) for fully connected multi-layer neural networks, using Neural Tangent Kernel theory to provide the excess population risk estimates for large sample sizes and network widths.

**Audience:**

Yes

**Broader Impact Concerns:**

The paper study the theoretical properties of differentially private training neural network. There is not any broader impact concern involved.

**Claims And Evidence:**

Yes

**Requested Changes:**

Please see the weakness part and below minor points.
### Minor Points

1. In Remark 9, the paper compares Theorem 9 and Theorem 1. Should it be comparing Theorem 9 with Theorem 2 instead?

2. Figures 4 and 5 plot the loss with respect to $T$ and $n$. How does this relate to the privacy budget? How is the privacy loss accumulated in these scenarios?

3. non-monotone decreasing should be monotone non-decreasing?

Overall, I would not recommend the publication of the paper.

**Strengths And Weaknesses:**

### Strong Points

1. This paper provides a comprehensive overview of the excess risk associated with differential privacy (DP) algorithms for neural networks. It includes analyses of Generalized Linear Models (GLMs) under both well-specified and misspecified models, two-layer neural networks with sigmoid and ReLU activation functions, and multi-layer neural networks within the Neural Tangent Kernel (NTK) framework.

2. The results presented in the paper appear to be correct, although I have not yet extensively verified them.

### Weak Points

1. The paper studies excess risk by transforming the non-convex optimization problem into a convex surrogate objective and using DP-enforced multi-phase SGD to optimize the surrogate. This approach is problematic for studying neural networks with DP, as it undermines the benefits of non-convexity or non-linearity that better fit the data. As a result, the achievable bounds do not provide much insight or guidance for the real problem of optimizing neural networks with DP-SGD.

2. Consequently, there is a huge gap between the theory presented in the paper and the best of practice. When training NN with DP guarantee, people use DP-SGD and clip individual gradients to control the sensitivity. This gives us a reasonable accuracy drop with decent privacy budgets. It does not make assumption on the objectives and network structures, which helps leveraging the advantage of sota NNs developed in practice.

2. The technological contribution is weak. There is no significant technological innovation in the proofs presented in the paper. Moreover, the paper does not discuss the lower bound, leaving the tightness of the upper bound and the potential for improvement unclear. The bounds presented are not advantageous compared to existing work, such as the findings in (https://arxiv.org/pdf/2006.06783). Practically, it would make more sense to exploit the problem's structure (low-rank or sparse structure) rather than relying solely on typical assumptions.

3. In the NTK regime, the approximation error is not accounted for, which is particularly important under the assumption ($n > O(m)$). More importantly, such a bound does not align well with practical observations, as evidenced by findings from the recent paper (https://arxiv.org/abs/2204.13650) that suggest the model's performance is not severely constrained by its width.

---

### Review · Reviewer_UmBT · 2024-07-03

**Summary Of Contributions:**

This paper studies differential private stochastic optimization problems for generalized linear models (GLMs) and neural networks with one or more hidden layers.

- For GLMs, the paper analyzed a phased SGD algorithm with Gaussian mechanism output perturbation, along with (an optional) random projection step to reduce the problem dimension.

- The paper also extends the analysis in GLMs to non-smooth link functions using Moreau evenlope, as well as the ReLu function (with a specialized analysis).

- For mis-specified GLMs, the paper studies isotropic log-concave distributions by borrowing the ideas from Diakonikolas et al., (2020), and establishes similar excess risk bounds for a private phased SGD algorithm with adaptively chosen noise variance.

- For one-hidden-layer neural networks, the paper combines results for GLMs with existing work on neural network approximation in RKHSs, and establishes excess risk bounds with privacy guarantees.

- For multi-layer neural networks, the paper studies a clipped projected private SGD using Gaussian mechanism gradient perturbation, and anallyzed its empirical loss in the NTK regime.

**Audience:**

Yes

**Broader Impact Concerns:**

The paper does not involve any ethical concerns or issues.

**Claims And Evidence:**

Yes

**Requested Changes:**

- In Section 4.1.2, there seems to be something wrong with the parameter $\gamma$. It is used to choose the iteration length $T$. In Theorem 4, it is chosen as $1/(n \log n)$, making the parameter $T$ of order $n^2 \log^2 n$, while the total sample size is just $n$ and we don't have enough data. In Theorem 5 it is chosen as $n \log n$, making the parameter $T$ much smaller than 1.

- Algorithm 4 is not readable. First, how is $w_t$ initialized in the iteration. Is it the case that $w_1 = w$? Is $w_t$ a vector or a scalar? (It seems that it needs to be a scalar, but the notation is very confusing as we have $w_t^i$ for model parameter in Algorithm 5). Furthermore, in line 7, $x_t$ is used before defined.

- In Theorem 4 and 5, the choices of $\beta$ and $\eta$ are nested. It is important to explain clearly how they are chosen. In particular, we need $\eta < \frac{2}{\beta}$ for some $\beta = O (\sqrt{n})$, while $\eta$ itself is bounded by $O (1 / \sqrt{n})$. We need to make sure that when constant factors are taken into account, the feasible region of $(\eta, \beta)$ contain a pair such that $\eta < \frac{2}{\beta}$. Otherwise the Theorems are vacuous.

- Assumption 2 in its current form can never be satisfied. In particular, the covariance of an isotropic distribution supported on the region $\{x: \| x\|_2 \leq \sqrt{d}\}$ is at most $I_d$, with equality holding true for a uniform distribution on sphere (which is not log-concave). I think the author means $\| x_i \|_2 \leq O( \sqrt{d} )$ instead of $\sqrt{d}$. But this needs to be checked.

- The $\ell^2$-norm of data and/or parameter may hide dimension dependence. For example, if the data are isotropic and bounded by $O (1)$, and the parameter norm is bounded by $O (1)$ as well, with the logistic activation, the output will be highly concentrated around $1/2$, which is not useful. A more natural scaling has data norm bounded by $O (\sqrt{d})$ (for example, a truncated standard Gaussian).

- Theorem 8. Unexpected line break in the equation.

- Theorem 13. What is $q$? Should it be $p$ or something?

**Strengths And Weaknesses:**

The paper contains several theoretical results. Some of them are novel while some others involve less new ideas. Here's a detailed discussion.

- The algorithm and theoretical analysis in Section 4.1 seem very similar to Arora et al., (2022). Indeed, I think (under some minor technical modifications), the results can be directly obtained from that paper. The author argued that Arora et al., (2022) requires non-negative loss while the current paper does not. This is not very convincing as the output is bounded and we can always apply a constant shift.

- Section 4.1.2 is poorly written, with numerous typos and unexplained issues that hinder the readers to understand the results (see detailed comments below).

- Section 4.2 makes use of some existing literature on GLMs with log-concave distributions. Though the proof idea is not new and the private SGD analysis is rather routine, it is still interesting to see a solution to private GLM in the mis-specified setting

- Section 5 is basically a direct combination of previous results with existing literature on neural network approximation.

- Section 6 takes a different route, by analyzing an existing private SGD algorithm that involves clipping and projection. The algorithm uses a sub-sampling scheme, with the data size actually used being much smaller than the number of hidden units (therefore in the over-parametrized scheme), though the actual sample size needs to be much larger than the number of hidden units and the majority of samples are wasted. This leads to a slow convergence rate in terms of $n$ (note that $m > T^7$ and $n > m \sqrt{T}$). To my knowledge, despite the weak results and the proofs being borrowed mostly from existing papers, this is the first end-to-end analysis of private SGD learning in the NTK regime.

Additionally, the rationale of the paper looks a bit confusing. The title emphasizes on ``Multi-layer Neural Networks'', while the majority of the text focuses on GLMs and their applications to one-hidden-layer neural networks. The multi-layer part looks like an additional result which is relatively disconnected from the main results. The authors may need to make it clearer what the main contributions are, and why they are novel. If each of the results could not qualify a TMLR-level journal, the collection of them does not either.

---

### Note · Authors · 2024-07-16

I have read and agree with the venue's withdrawal policy on behalf of myself and my co-authors.